# 3DGS Is A Versatile Regulator: Modulating Universal Metric-depth Representation via Anchor-based Gaussian-Splatted Multiplication

## Abstract

Recent advances in zero-shot affine-invariant depth estimation have achieved remarkable progress. However, extending relative depth to metric depth remains challenging due to the absence of reliable metric-scale guidance within existing depth foundation models. Building on this, we introduce a novel depth estimation paradigm—*anchor–multiplier factorization*—as an alternative to conventional approaches such as direct depth regression, depth completion, or feature-fusion methods. Our key insight is that sparse point anchors supply indispensable metric-scale cues, while relative-scale geometric structure can be stably regulated via Gaussian-splatted multiplication conditioned on image semantics. Accordingly, we implement GSD—an anchor-based Gaussian Splatting Depth Regulator for universal metric-depth restoration. We also propose *the first* theoretical analysis showing how anchor–multiplier factorization mitigates training divergence, and thereby improves metric restoration accuracy. Extensive experiments across diverse datasets demonstrate substantial accuracy gains over state-of-the-art baselines, highlighting the benefits of *treating 3DGS not merely as a renderer, but as a versatile regulator* for visual representation learning.

## 1 INTRODUCTION

Universal depth estimation plays a critical role in 3D vision, enabling myriad downstream applications in 3D reconstruction, autonomous driving, and robotics. Current monocular foundation models (Yang et al., 2024b; Ke et al., 2024) have reached a *Eureka* moment in zero-shot affine-invariant relative depth estimation, which can produce high-resolution and well-structured depth maps for wild images. However, bridging the gap from relative depth to metric depth remains challenging. Fine-tuning these models on metric depth data often causes **catastrophic forgetting** of previously learned relative geometry. As the metric-scale supervision loss converges, previously sharp depth boundaries become blurred, and the overall depth generalization capability deteriorates.

Prior efforts to achieve universal depth estimation have primarily followed three major pipelines: 1) direct depth regression with large transformer (Yang et al., 2024a; Hu et al., 2024) or diffusion models (Ke et al., 2024; Guizilini et al., 2025) ; 2) geometry estimation guided by semantic or multi-modal prompts (Wang et al., 2025a;b; Fu et al., 2024); and 3) post-processing or explicit scaling (e.g., 3D Gaussian Splatting (3DGS) for self-supervised refinement or least-squares scale alignment (Charatan et al., 2024; Xu et al., 2025)). While these methods generally focus on affine-invariant relative depth, some of the expansion methods (Yang et al., 2024b; Lin et al., 2025; Viola et al., 2024) incorporate information from sparse anchors to restore the metric depth in the real scale(Figure 1). However, hey still suffer from geometric degradation and knowledge forgetting, even with sparse depth anchors provided (e.g., comparing Depth Anything v2 + Least Squares post processing [DAv2+LS] and finetuned Depth Anything v2 Metric depth [DAv2 Metric] in Table 2). This raises a fundamental question: *can we establish a theoretically grounded paradigm leveraging sparse 3D cues to convert universal relative depth predictions into metric scale?*

We address this question in the affirmative by introducing **anchor–multiplier factorization** (illustrated in Figure 1 and Figure 3). In this novel paradigm, metric-scale depth is estimated as the

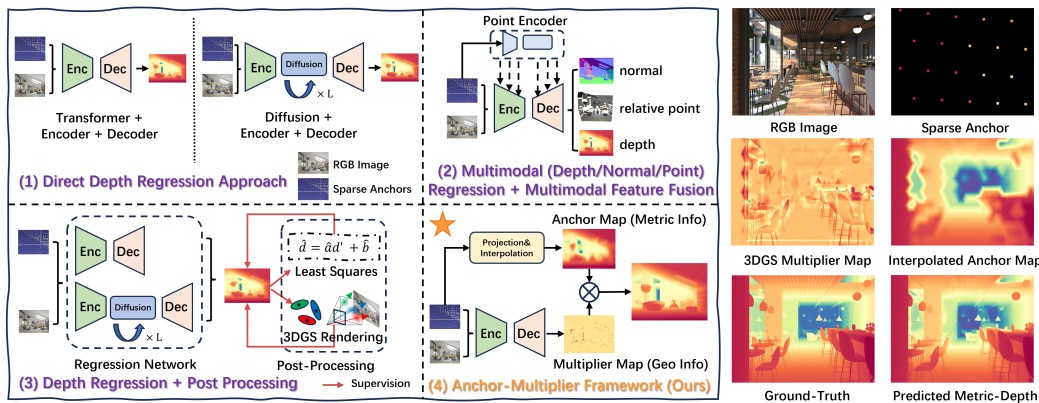

Figure 1: **Overview of the Anchor-Multiplier Paradigm.** The left part illustrates the differences between our paradigm and the previous three pipelines. The right part displays the visualization of our 3DGS multiplier map, interpolated anchor map, and the factored result of high-accuracy predicted metric-depth.

product of an interpolated sparse-anchor depth map (providing coarse metric scale) and a learnable dense multiplier map (refining relative geometry details). This decoupling of metric scale and relative geometric structure largely alleviates the "mutual cancellation" effect that causes spatial knowledge to be forgotten. Although some prior research (Yin et al., 2023; Piccinelli et al., 2024; Wang et al., 2025c) has also noticed such benefits, they undertake this problem via outside camera metric information injection (Piccinelli et al., 2024), field-of-view augmentation (Saxena et al., 2023), canonical 3D space design (Yin et al., 2023), or extra scale estimation networks Wang et al. (2025c) *etc*. We observe that these strategies partially disentangle scale from structure but do not fully resolve the issue. The primary goal of our anchor-multiplier schema is to learn the per-pixel multiplier map that stretches or compresses the coarsely interpolated anchor map to align with the ground truth, which is the underlying difference compared to the three paradigms above. We further provide **a theoretical analysis** showing that learning the multiplier is statistically easier and yields more stable gradients than direct depth regression (Section 3.2).

To implement the multiplier, we repurpose 3D Gaussian Splatting (3DGS) as a regulator (Section 4). In previous work, 3DGS is utilized through its Gaussian primitives (each with a 3D center, covariance, opacity, etc.) to render RGB images, yet we adopt 3DGS feature representation here to produce the multiplier map for coarse depth regulation. This choice is motivated by 3DGS's ability to capture rich spatial textures—orientation, scale, transparency—beyond what point or voxel representations can offer. After training, the multiplier maps delineate semantic boundaries (see Figure 1) and a relative geometric structure, remarkably resembling the output of dedicated relative-depth foundation models, demonstrating 3DGS's role as a versatile regulator for depth representation.

Our contribution can be summarized as follows: 1) We propose a novel **anchor-multiplier paradigm that cleanly disentangles** the relative geometry from metric-scale information; 2) We provide **the first theoretical analysis** of this factorization, proving that multiplier learning is statistically easier and stabilizes training; 3) We introduce GSD, the first framework to **employ a 3DGS representation as a depth regulator** and achieves geometric fidelity competitive with depth foundation models. 4) Extensive quantitative and qualitative experiments on diverse benchmarks show significant improvements over state-of-the-art baselines, validating the effectiveness of our innovative approach.

## 2 RELATED WORK

Due to space constraints, we will briefly discuss the technical roadmap and emphasize its representative features here. A detailed literature review is described in the Appendix A.1.

## 2.1 Monocular Depth Estimation and Depth Completion

Monocular depth estimation aims to predict per-pixel depth from a single image, whereas depth completion refines this prediction using sparse anchor depth cues. Earlier work typically employs task-specific modules, including confidence-based propagation (Park et al., 2020; Tang et al., 2024), multi-resolution geometric priors (Bartolomei et al., 2024; Zuo et al., 2024), or sparse cues operator-level enhancement (Conti et al., 2023; Zhang et al., 2023). Recent efforts converge on foundation-scale models pursued along four axes: 1) camera-aware representations that resolve scale ambiguity (Yin et al., 2023; Hu et al., 2024; Piccinelli et al., 2024), 2) transformer (Yang et al., 2024a;b; Lin et al., 2025) or diffusion (Guizilini et al., 2025; Ke et al., 2024; Fu et al., 2024; Viola et al., 2024) backbones trained for affine-invariant prediction, 3) boundary-aware gradient losses design (Bochkovskii et al., 2024) and specific temporal matching architectures (Gui et al., 2025) tuned for fidelity or speed, and 4) multi-task geometry estimation frameworks that jointly optimize depth with surface normals or point maps (Wang et al., 2025a;b; Keetha et al., 2025). In contrast, our anchor–multiplier distinguishes itself from the formulations above by explicitly factoring the metric scale from relative geometry, rather than relying on regressing metric depth.

## 2.2 Depth Estimation with 3DGS

Contemporary 3D Gaussian Splatting (3DGS) increasingly leverages monocular depth to stabilize scale and sharpen geometry for rendering, including depth-conditioned initialization or supervision (Xu et al., 2025; Chung et al., 2024; Zheng et al., 2025), depth confidence thresholding and multi-cue fusion (Zhang et al., 2025; Deng et al., 2025), geometric regularization from relative normal cues for consistent novel-view synthesis (Zhan et al., 2025; Hu et al., 2025; Lee et al., 2024), and *etc*. To compare with, our Gaussian Splatted Depth (GSD) network applies 3DGS as a constraint coefficient under our anchor-multiplier paradigm, instead of explicitly leveraging its rendering ability for regression metric-value.

# 3 Problem Setup and Theoretical Justification

## 3.1 Notation and Setup

Given an image sample with pixel coordinates $\mathbf{I} = I(u, v)$ and ground-truth depth $D_{\text{gt}} = D_{\text{gt}}(u, v)$, conventional depth regression directly predicts depth via $\hat{D} = f_\theta(\mathbf{I})$, where $f_\theta$ denotes a depth regression network parameterized by $\theta$. Depth completion can be viewed as an extension of depth regression, parameterized as $\hat{D} = f_\theta(\mathbf{I}, S)$, where $S \in \{(u, v)\}$ is defined as the projection of a set of anchor points carrying metric depths, resulting in a sparse depth map where unprojected areas are void. The general loss and its gradient are then

$$\mathcal{L} = \text{Loss}(f_\theta(\mathbf{I}, S), D_{\text{gt}}), \tag{1}$$

$$g_\theta^{\text{trad}} = \frac{\partial \mathcal{L}}{\partial \theta} = \frac{\partial \mathcal{L}}{\partial f} \cdot \frac{\partial f}{\partial \theta}, \tag{2}$$

where $g_\theta^{\text{trad}}$ denotes the **g**radient with respect to the **trad**tional methods for model parameter $\theta$.

In contrast, we propose an anchor–multiplier factorization of depth completion that rewrites $\hat{D}$ as

$$\hat{D} = f_\theta(\mathbf{I}, S) = \hat{\alpha} \cdot \mathcal{I}(S), \tag{3}$$

$$\hat{\alpha} = \alpha_\theta(\mathbf{I}, S), \tag{4}$$

where $\mathcal{I}(S)$ is a dense scalar matrix obtained by interpolating $S$ to fill void values, and $\hat{\alpha}$ is a pixel-wise scale multiplier that modulates $\mathcal{I}(S)$ to match $D_{\text{gt}}$. **Intuitively, $\mathcal{I}(S)$ provides a coarse global metric scale, while $\hat{\alpha}$ captures fine-grained, affine-invariant local geometry (See visualizations of $\mathcal{I}(S)$ and $\alpha_\theta$ in Figure 3).** In practice, $\hat{\alpha}$ is produced by a neural network $\alpha_\theta(\cdot)$. The resulting gradient becomes

$$g_\theta^{\text{new}} = \frac{\partial \mathcal{L}}{\partial \theta} = \mathcal{I}(S) \cdot \frac{\partial \mathcal{L}}{\partial f} \cdot \frac{\partial \alpha}{\partial \theta}. \tag{5}$$

It can be observed that $\frac{\partial \mathcal{L}}{\partial f}$ denotes supervision-driven derivatives with respect to prediction errors, while $\frac{\partial \alpha}{\partial \theta}$ reflects model sensitivity derivatives with respect to image input. In between, $\mathcal{I}(S)$ acts as a known sample-dependent constant that splits the gradient $g_\theta^{\text{new}}$ into a product of a triplet and reduces the correlation degree of the $\frac{\partial \mathcal{L}}{\partial f}$ and $\frac{\partial \alpha}{\partial \theta}$ (See Assumption 3).

### 3.2 GRADIENT STABILITY AND THEORETICAL JUSTIFICATION

Let $U = \frac{\partial \mathcal{L}}{\partial f}$, $V = \frac{\partial f}{\partial \theta}$, and $W = \frac{\partial \alpha}{\partial \theta}$, we have

**Assumption 1** (Bounded Multiplier Assumption). *Given an image sample* **I** *with sparse anchors S, the function $\alpha_\theta$ is usually bounded and smooth over the continuous depth domain. Specifically, we assume that $\alpha_\theta$ admits a Lipchitz-type boundary: $0 < \alpha_{min} \leq \alpha_\theta(\mathbf{I}, S) \leq \alpha_{max} < \infty$, and there exists $0 < \kappa < \infty$ such that $||W|| \leq \kappa$.*

**Assumption 2** (Metric-Depth Variation Assumption). *For the traditional direct depth regression approach, the magnitude $V = \frac{\partial f}{\partial \theta}$ largely increases due to the large variations of metric-depth scales across various scenes (e.g. indoor v.s. outdoor scenery). Specifically, there exists $0 < \Lambda < \infty$ such that $||V|| \geq \Lambda$.*

**Assumption 3** (Weak Dependence Assumption). *We assume $U$ (supervision-driven gradients) and $V$ or $W$ (model-sensitivity gradients) are weakly correlated, i.e., $|\text{Corr}(U, V)| \to 0$ or $|\text{Corr}(U, W)| \to 0$.*

**Theorem 1** (Variance Reduction Theorem, VRT). *Under Assumptions 1–3, if for some $\kappa, \Lambda < \infty$,*

$$\mathbb{E}[||W^2||] \leq \kappa^2 \quad and \quad \mathbb{E}[||V^2||] \geq \Lambda^2, \tag{6}$$

*then*

$$\mathbb{E}\left[\frac{\text{Var}[g_\theta^{new}]}{\text{Var}[g_\theta^{trad}]}\right] \lesssim \mathbb{E}[\mathcal{I}(S)^2] \cdot \frac{\kappa^2}{\Lambda^2}, \tag{7}$$

where $\mathbb{E}(\cdot)$ abbreviates the statistical expectations over the whole dataset samples. A detailed proof is provided in the Appendix Section A.2. Especially, under mild conditions $\kappa \ll \Lambda \Rightarrow \mathbb{E}\left[\frac{\text{Var}[g_\theta^{\text{new}}]}{\text{Var}[g_\theta^{\text{trad}}]}\right] < 1$, it indicates that factorization can give a more constrained and stable gradient, which improves model convergence. Moreover, our factorization formulation also explicitly injects 3D prior information, $\mathcal{I}(S)$, directly into back-propagated gradients via equation 5.

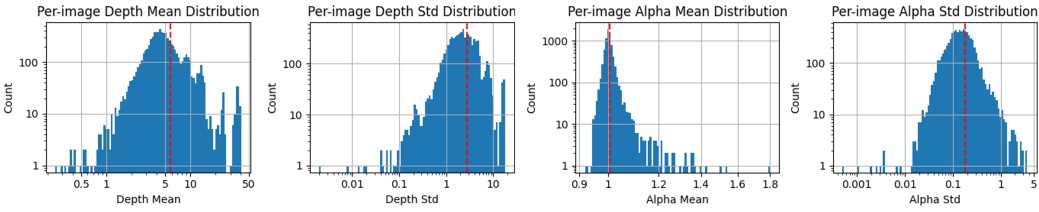

Figure 2: **Histogram comparison of per-image means and standard deviations for depth** $D$ **vs. multiplier** $\alpha$**.** Left two images: mean and std distributions for $D$; right two images: mean and std distributions for $\alpha$. The red dashed line shows the dataset-wide expectation.

As illustrated in Figure 2, statistical results on Hyersim dataset (Roberts et al., 2021) also support our proposed anchor–multiplier VRT (Theorem 1). We compute, per each image, the mean and standard deviation of the depth ground truth $D$ and of $\alpha$, and then plot the resulting histograms over the entire dataset. The distributed ranges for $\alpha$ (Mean$[\alpha] \lesssim 3.5$, Var$[\alpha] \lesssim 8.5$) are markedly smaller and tighter than those for depth (Mean$[D] \lesssim 50.4$, Var$[D] \lesssim 17.8$), indicating that learning $\alpha$ is statistically easier than directly regressing $D$.

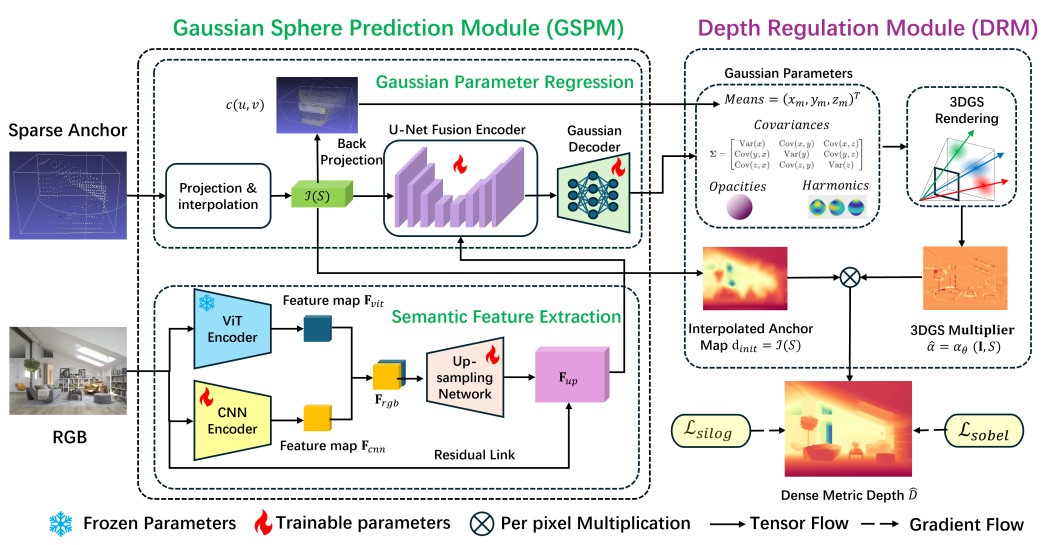

Figure 3: **Overview of the *GSD* Architecture.** Following the Anchor-Multiplier design merits, we adopt 3DGS as the multiplier regressor $\alpha_\theta$. The whole network can be decomposed into two modules: the Gaussian Sphere Prediction Module (GSPM) and the Depth Regulation Module (DRM). GSPM takes an RGB image and sparse anchors to form the interpolated anchor map $\mathcal{I}(S)$ and predict 3DGS parameters. Then, DRM is designed to splat gaussian spheres into a multiplier map $\alpha_\theta$. Finally, multiplying $\alpha_\theta$ by $\mathcal{I}(S)$ yields the final dense metric depth $\hat{D}$.

## 4 METHODOLOGY

### 4.1 OVERALL FRAMEWORK

The overall architecture is exhibited in Figure 3. We instantiate the multiplier network $\alpha_\theta$ with 3D Gaussian Splatting (3DGS) to modulate/edit the interpolated anchor map $\mathcal{I}(S)$ into metric depth $\hat{D}$, because 3DGS can provide strong alignment to semantic textures, while jointly incorporating representation from the RGB image and the 3D anchor prior. The whole network factorizes into two modules: the **Gaussian Sphere Prediction Module (GSPM)** and the **Depth Regulation Module (DRM)**. GSPM comprises a semantic feature extractor and a Gaussian-parameter regressor. It first encodes the RGB image and sparse depth anchors into features with residual, then regresses Gaussian parameters: sphere centers (carrying implicit depth), covariances (defining each sphere's spatial support), opacities, and spherical harmonics (SH) coefficients (controlling local multiplicative value). Based on predicted Gaussian parameters, DRM performs differentiable rendering to produce a dense multiplier map, which is applied to $\mathcal{I}(S)$ to obtain the final metric prediction $\hat{D}$.

### 4.2 GAUSSIAN SPHERE PARAMETERS PREDICTION

**Sparse anchor projection & interpolation.** Sparse depth anchors may come from LiDAR, RGB-D sensors, SLAM, sparsification ground truth, or simulated LiDAR. Given a sparse anchor point set $\{P_i \in \mathbb{R}^4\}$ (represented by quaternion), camera extrinsic matrix $E \in \mathbb{R}^{4 \times 4}$ and intrinsic matrix $K \in \mathbb{R}^{3 \times 4}$, we compute the interpolated depth map initialization $d_{init}$ at the resolution of $\mathbf{I} \in \mathbb{R}^{H \times W}$,

$$S = KEP_i \tag{8}$$
$$d_{init} = \mathcal{I}(S, H, W). \tag{9}$$

For the interpolation function $\mathcal{I}$, we adopt a combined strategy: linear interpolation for inner values and nearest-neighbor interpolation for the boundary.

**Initial depth back-projection.**

Using the pinhole camera model, the homogeneous pixel coordinates $(u, v, 1)^T$ are back-projected to ray vectors under the camera coordinate system:

$$ray(u, v) = K^{-1}(u, v, 1)^T. \tag{10}$$

So the mean position $p_{u,v}$ of 3D Gaussian kernel corresponding to each pixel in the camera frame is

$$p_{u,v} = d_{init} \cdot ray(u, v) = d_{init} \cdot K^{-1}(u, v, 1)^T. \tag{11}$$

Through such a back-projection procedure, we have the means of all 3D Gaussian spheres.

**Semantic feature extraction.** We design to let our feature extractors have both global awareness (spatial layouts and geometric structure) and local awareness (boundary and texture). Therefore, we adopt a ViT-CNN hybrid encoder to capture both scene-level context and fine-grained features. The ViT branch inherits the frozen weights of the pretrained DINOv2 to provide global semantics and long-range dependencies, whereas the other branch employs a learnable convolution network and focuses on localized cues. The concatenated features $\mathbf{F}_{rgb}$ from two branches are upsampled to obtain $\mathbf{F}_{up}$ to align with the resolution of $\mathbf{I}$, and then passed to the U-Net fusion encoder.

**Gaussian parameter regression.** Following MVSplat (Chen et al., 2024), we utilize a U-Net for multimodal fusion and refinement. The inputs—$\mathbf{F}_{up}$, raw image input $\mathbf{I}$, interpolated anchor depth map $d_{init}$, and the anchor mask $M_{anchor}$—are fused with residual connections:

$$\mathbf{F}_{\text{refined}} = \text{UNet}(F_{up}^{\theta}, \mathbf{I}, d_{init}, M_{anchor}). \tag{12}$$

Subsequently, a Gaussian decoder transforms $\mathbf{F}_{\text{refined}}$ to Gaussian parameters:

$$\mathbf{G} \doteq (\mathbf{G}_{\text{means}}, \mathbf{G}_{\text{covariances}}, \mathbf{G}_{\text{harmonics}}, \mathbf{G}_{\text{opacities}}), \tag{13}$$

$$\mathbf{G}_{\text{means}} = c(u, v), \tag{14}$$

$$\{\mathbf{G}_{\text{covariances}}, \mathbf{G}_{\text{harmonics}}, \mathbf{G}_{\text{opacities}}\} = \text{GS-Decoder}(\mathbf{F}_{\text{refined}}). \tag{15}$$

With these Gaussian parameters, the following splatting procedure proceeds analogously to RGB-space rendering.

## 4.3 DEPTH REGULATION MODULE

**Gaussian-splatting rendering.**

Remember that we use 3DGS as a regulator to render the multiplier $\alpha_\theta$. Following typical RGB rendering practice, we adopt a pixel-wise feed-forward parallel differentiable rendering pipeline

$$\alpha_{\text{res}}^{\theta} = \text{Feed-Forward Render}(\mathbf{G}), \tag{16}$$

since feed-forward splatting approach eliminates the need for lengthy iterative optimization and enables a much faster training/inference speed in an end-to-end manner. Gaussian primitives are first rasterized using their 3D means, covariances (represented by upper triangular elements), opacities, and spherical harmonics (SH) coefficients. The rasterization process produces a rendered depth image for each Gaussian, which is accumulated to obtain the rendered multiplication factor $\alpha_{res}^{\theta}$.

**Metric multiplier transformation.** Although we intend for the output $\alpha_{\text{res}}^{\theta}$ to serve directly as a multiplicative factor, it is observed that the rendered result has a limited numerical range, *i.e.* $\alpha_{\text{res}}^{\theta} \in [0, 1]$, due to the Gaussian Rasterization property with SH and normalization pipeline. To compare with, the real multiplier $\alpha_{\text{gt}}$ value is defined by the ratio of the ground-truth metric depth $D_{\text{gt}}$ and the interpolated anchor depth map $d_{init}$ from Equation (3)

$$\alpha_{\text{gt}} \doteq \frac{D_{\text{gt}}}{d_{init}}, \tag{17}$$

which may have a different numerical range: $\alpha_{\text{gt}} \in [0, \alpha_{max}], \alpha_{max} \neq 1$. Therefore, we need to re-map $\alpha_{\text{res}}^{\theta}$ to a new value range $[0, \alpha_{max}]$. We note that the Gaussian Splatting pipeline responds

with 0.5 uniformly to the initial all-zero parameter input so that when $\alpha_{\text{res}}^{\theta} = 0.5$, the actual derived multiplier should be $\alpha_{\theta} = \phi(\alpha_{\text{res}}^{\theta}) = 1$, which keeps an identity transformation and does not compress or stretch the $d_{\text{init}}$. Therefore, we define $\phi(\cdot)$ as

$$\alpha_{\theta} = \phi(\alpha_{\text{res}}^{\theta}) \doteq \begin{cases} \dfrac{(\alpha_{\max} - 1)^{2 \cdot \alpha_{\text{res}}^{\theta}} - 1}{\alpha_{\max} - 2}, & \text{if } \alpha_{\max} > 2 \\ 2 \cdot \alpha_{\text{res}}^{\theta}, & \alpha_{\max} \leq 2, \end{cases} \tag{18}$$

subject to $\phi(0) = 0$, $\phi(0.5) = 1$, and $\phi(1) \geq \alpha_{\max}$.

Finally, we obtain the dense metric prediction via pixel-wise multiplication:

$$\hat{D} = d_{init} \otimes \alpha_{\theta}. \tag{19}$$

### 4.4 LOSS FUNCTION AND TRAINING PARADIGM

**Loss Function.** Following previous practice (Eigen et al., 2014), we adopt the Scale-Invariant Logarithmic (SILog) loss for depth regression. Interestingly, we find that the original SILog loss supervision over metric-depth has a close connection with the supervision over the multiplier $\alpha$

$$r_{u,v} = \log \hat{D} - \log D_{gt} = \log \frac{d_{init} \cdot \alpha_{\theta}}{D_{gt}} = \log \frac{\alpha_{\theta}}{\alpha_{\text{gt}}}, \tag{20}$$

$$L_{SILog} = \overline{r^2}_{u,v} - \lambda \cdot (\overline{r}_{u,v})^2, \tag{21}$$

where $\lambda = 0.5$. Therefore, by applying SILog loss over $\hat{D}$, we implicitly supervise $\alpha_{\theta}$ with $\alpha_{\text{gt}}$.

For synthetic datasets, such as Hypersim and Virtual KITTI (Cabon et al., 2020), which have dense ground-truth labels, we adopt the Sobel operator-based gradient loss to enhance the edge sharpness

$$\nabla_x D = \text{conv}(D, F_x), \quad \nabla_y D = \text{conv}(D, F_y), \tag{22}$$

$$L_{Sobel} = \text{AVERAGE}_{u,v}(|\nabla_x \hat{D} - \nabla_x D| + |\nabla_y \hat{D} - \nabla_y D|), \tag{23}$$

where $F_x$ and $F_y$ are Sobel Kernels. As for real datasets that only have sparse annotations, we can also utilize gradient loss by first scaling the predicted depth to an affine-invariant scale, and then supervising it with pseudo labels generated by affine-invariant depth foundation models.

Finally, our training loss formula is

$$L = L_{SILog} + 0.1 \cdot L_{Sobel}. \tag{24}$$

## 5 EXPERIMENTS

### 5.1 DATASETS AND EXPERIMENTAL SETUP

We train our GSD model on the Hypersim (Roberts et al., 2021) training set and evaluate the in-domain results on the validation set. Following previous works (Viola et al., 2024; Lin et al., 2025; Liu et al., 2024b), we sample anchor points from the ground truth depth to generate LiDAR-like sparse depth input under several stride settings mentioned Section A.3. We also train GSD from scratch on the KITTI Completion dataset (Geiger et al., 2013)—a real-world driving scene dataset with paired RGB images and sparse LiDAR depth.

We evaluate our GSD model in a zero-shot manner on five unseen real-world datasets NYUV2(Silberman et al., 2012), KITTI(Geiger et al., 2013), Scannet(Dai et al., 2017), ETH3DSchops et al. (2017) and DIODE(Vasiljevic et al., 2019). The training configurations and detailed datasets description are illustrated in Section A.3. It is worth noting that the Hypersim training set is a indoor synthetic dataset with only 59k training samples.

We also trained GSD from scratch on the KITTI Completion dataset—a real driving-scene dataset with paired RGB images and sparse LiDAR depth. Its semi-dense ground truth is derived from temporal accumulation of consecutive LiDAR frames. For outdoor zero-shot setting we trained GSD on Virtual KITTI dataset with about 21k samples.

## 5.2 QUANTITATIVE COMPARISON

**Comparison methods.** We compare our method to various baselines as shown in Table 1 and Table 2. All the methods have been trained on Hypersim (Roberts et al., 2021), and we have fine-tuned them on KITTI (Geiger et al., 2013) for a fair comparison.

The post fusion method of Depth Anything v2 refers to scale and shift based least squares alignment with relative depth prediction. We follow the official codebase to fine-tune Depth Anything V2 for metric depth estimation on their provided checkpoints for indoor and outdoor scenes respectively, as DAv2 Metric.

According to Prompting Depth Anything, their released checkpoint is pretrained on Hypersim and then other two datasets, and we reproduce the training process of PromptDA according to its paper, which achieves a much better result. The detailed evaluation protocol can be found in Section A.3.

Table 1: Quantitative comparison on the Hypersim dataset and the KITTI Completion dataset of in-domain metric depth completion. All of the methods have been pretrained on Hypersim. Methods marked with * are finetuned with their released models and code on KITTI. All metrics are presented in percentage terms, and the unit of RMSE is meters.

| | dataset | | Hypersim val | | | | KITTI completion val | | |
| | anchor stride/prompt res. | | $7, 192 \times 256$ | | $16, 518 \times 686$ | | —, $378 \times 1246$ | | |
| Method | fusion method | retrained | AbsRel↓ | $\delta_1$ ↑ | AbsRel↓ | $\delta_1$ ↑ | AbsRel↓ | Rmse↓ | $\delta_1$ ↑ |
|---|---|---|---|---|---|---|---|---|---|
| DAv2+LS (Yang et al., 2024b) | post | no | 24.9 | 67.6 | 25.3 | 67.1 | 27.9 | — | — |
| DAv2 Metric* (Yang et al., 2024b) | post | yes | 18.6 | 75.2 | 19.0 | 74.8 | 10.9 | 3.846 | 89.8 |
| Marigold-DC (Viola et al., 2024) | model | optimize | 8.0 | 97.1 | 7.1 | 97.1 | 4.9 | 1.823 | 97.4 |
| PromptDA* (Lin et al., 2025) | model | yes | 3.1 | 98.1 | 3.0 | 98.1 | **2.0** | **1.229** | **99.1** |
| Ours(ViT-S) | GS regulate | yes | 3.0 | 97.7 | 3.0 | 97.7 | 2.4 | 1.667 | 98.1 |
| Ours(ViT-L) | GS regulate | yes | **1.7** | **98.5** | **1.7** | **98.6** | 2.0 | 1.348 | 98.7 |

**Qualitative comparison** As presented in Table 1, our GSD achieves competitive results under in-domain prediction compared with other depth estimation or completion baselines. Notably, our approach inherits the rich priors from the pre-trained image encoder and is solely trained on 59k synthetic samples. We believe that the excellent performance stems from the paradigm regulated by 3DGS. This aligns with our hypothesis that depth completion gains greater benefits from decoupling the global scale and the local geometric structure via sparse anchors and a residual multiplier.

Table 2: Zero-shot performance with stride of 16. Metrics in gray are from their original papers. Best results are **bold**, second-best are underlined.

| Method | NYUv2 | | KITTI | | ScanNet | | ETH3D | | DIODE | |
| | AbsRel↓ | Rmse↓ | AbsRel↓ | Rmse↓ | AbsRel↓ | Rmse↓ | AbsRel↓ | Rmse↓ | AbsRel↓ | Rmse↓ |
|---|---|---|---|---|---|---|---|---|---|---|
| NLSPN (Park et al., 2020) | — | 0.716 | — | 2.076 | — | 0.127 | — | — | — | — |
| SpAgNet (Conti et al., 2023) | — | 0.292 | — | 1.788 | — | — | — | — | — | — |
| CompletionFormer (Zhang et al., 2023) | — | 0.374 | — | 1.935 | — | 0.232 | — | — | — | — |
| VPP4DC (Bartolomei et al., 2024) | — | 0.247 | — | 1.609 | — | 0.076 | — | — | — | — |
| DepthSplat (Xu et al., 2025) | — | — | 10.7 | — | 3.8 | 0.144 | — | — | — | — |
| DepthLab (Liu et al., 2024b) | 2.5 | 0.276 | 7.2 | 2.171 | 2.3 | 0.081 | 3.1 | — | 17.6 | — |
| OMNI-DC (Zuo et al., 2024) | 2.3 | 0.225 | — | 2.058 | — | — | 5.3 | 1.069 | — | — |
| Depth prompting (Park et al., 2024) | — | 0.144 | — | 1.351 | — | — | — | — | — | — |
| DAv2+LS (Yang et al., 2024b) | 12.0 | 0.384 | 31.0 | 6.751 | 9.1 | 0.215 | 16.2 | 1.297 | 39.0 | 4.367 |
| DAv2 Metric (Yang et al., 2024b) | 5.6 | 0.206 | 4.5 | 1.861 | 21.2 | 0.406 | 30.4 | 2.278 | 44.1 | 7.827 |
| PromptDA (Lin et al., 2025) | 2.0 | 0.132 | **5.9** | 4.011 | 2.2 | 0.094 | 2.8 | **0.453** | 15.1 | **1.733** |
| Marigold-DC (Viola et al., 2024) | 1.9 | **0.119** | 10.6 | 3.575 | **1.6** | **0.079** | — | 2.008 | 14.4 | 2.659 |
| Ours | **1.8** | 0.130 | 6.4 | **3.570** | **1.6** | 0.086 | 2.6 | 0.503 | **13.1** | 2.560 |

The zero-shot performance with a stride of 16 is presented in Table 2. Our GSD attains the best balance in terms of accuracy and efficiency across these zero-shot scenes, highlighting the generalization ability of introducing a 3DGS regulator.

**Quantitative comparison** As shown in Figure 4 and Figure 5, we mark the anchor points with various strides and visualize the initial coarse depth. The error map is presented on the top-left while significant errors areas are highlighted with boxes and arrows. The result of our method demonstrates significantly better geometric coherence with the sparse inputs. More visualization comparisons are shown in Section A.5.

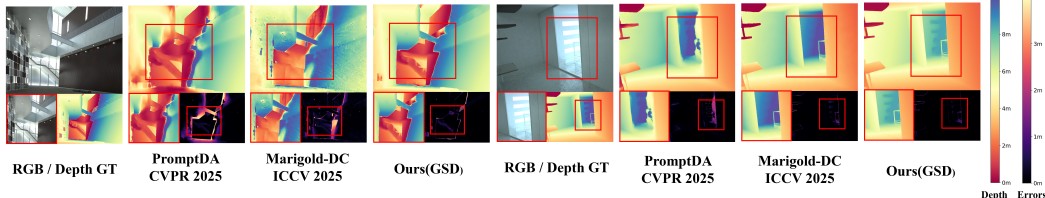

Figure 4: **Qualitative comparisons with the latest state-of-the-art methods.** PromptDA refers to Prompting Depth Anything (Lin et al., 2025), and Marigold-DC (Viola et al., 2024) denotes the depth completion variant of Marigold (Ke et al., 2024). We compare full-depth predictions, zoomed-in local regions, and corresponding error maps based on normalized absolute depth errors.

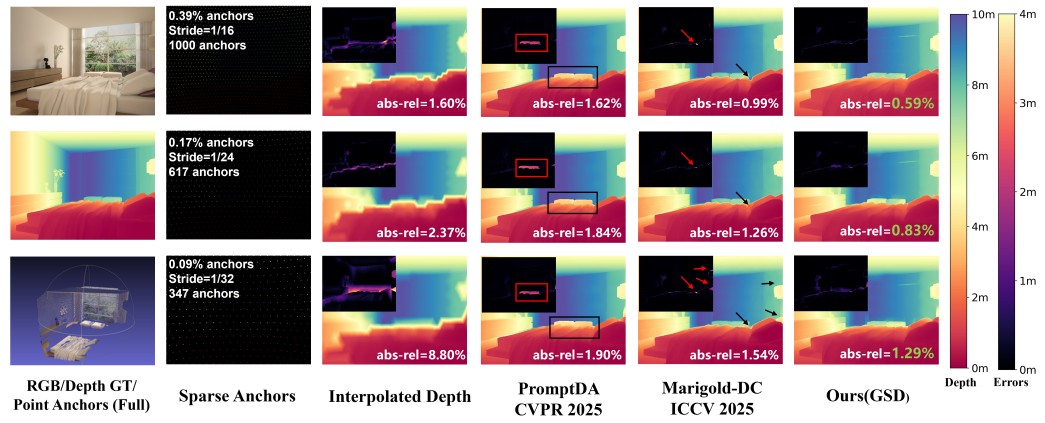

Figure 5: **Qualitative comparisons under varying anchor densities.** Normalized absolute depth-error maps are shown in the top-left of each panel. Across all density settings, our method consistently produces the most accurate results. Zoom in for better results.

## 5.3 ABLATION STUDIES

Table 3: Ablation of modules.

| modules | absrel | rmse | $\delta_1$ |
|---|---|---|---|
| (a) Interpolation | 4.3% | 0.621 | 0.961 |
| (b) a + predict metric | 3.6% | 0.566 | 0.970 |
| (c) b + UNet | 3.3% | 0.524 | 0.973 |
| (d) c + *multiplier | 3.1% | 0.517 | 0.974 |
| (e) d + GS decoder | 2.8% | 0.502 | 0.977 |

Table 4: Ablation on anchors.

| s | res | train | Interp rel | Interp rmse | GSD rel | GSD rmse |
|---|---|---|---|---|---|---|
| 7 | low | ✓ | 4.8 | 0.636 | 1.7 | 0.378 |
| 7 | high | ✓ | 2.6 | 0.468 | 1.4 | 0.336 |
| 16 | high | ✗ | 4.3 | 0.621 | 1.7 | 0.384 |
| 32 | high | ✗ | 6.9 | 0.796 | 2.9 | 0.493 |
| 64 | high | ✗ | 11.3 | 1.054 | 8.2 | 0.865 |

Table 5: Ablation of backbones.

| encoder | Hyersim absrel | Hyersim rmse | KITTI-DC absrel | KITTI-DC rmse |
|---|---|---|---|---|
| ViT-S | 3.4% | 0.530 | 2.4% | 1.667 |
| ViT-L | 3.2% | 0.495 | 2.0% | 1.348 |

Table 3 validates the effectiveness of our proposed modules and methods when evaluated with a stride of 32 at the resolution of $518 \times 686$ on Hypersim validation set. (a) shows the quantitative indicators through interpolation. (b) adds the semantic feature extraction module and mlp layers to predict the metric depth itself with training procedure, and (c) refines the features through UNet fusion module. In configuration (d) we use our anchor–multiplier factorization instead of regressing the metric depth directly. Consistent with our theory, the gradient of multiplier is smoother and it is the faster to achieve convergence while the performance has also improved. Finally in (d) we add the 3DGS regulator, and it shows that 3DGS beats an equally pure 2D $\alpha$-predictor and achieves the best performance.

Table 4 shows the performance of GSD with fewer metric clues. The resolution of sparse depth map $d_{init}$ is $256 \times 192$ for low and $686 \times 518$ for high, while the resolution of RGB image is $686 \times 518$. The ✓ refers that GSD is trained under the stride while using them for evaluation on others ✗ settings. The metric "rel" refers to absolute relative error expressed in percentage, and root mean square error is measured in meters.

Experiments in Table 5 are conducted under the resolution of $256 \times 192$ with a stride of 16, demonstrating the significance of high quality features. The total parameters of our model are 308 million(3.8 million trainable) for ViT-L and 25 million(3.1 million trainable) for ViT-S.

Our model architecture is highly flexible, allowing the combination of feature extraction backbone, (un)freezing encoders for semantic enhancement, such as replacing the ViT encoder with DINOv3 (Siméoni et al., 2025) or introducing another branch from SigLIP2 (Tschannen et al., 2025). Actually the trainable parameters of our GSD of ViT-L is only 3.8 million while the other 300 million parameters are frozen. Equipped with the prior injection of the image encoding module pretrained on large-scale real-world datasets, we expect to achieve better experimental results.

## 6    CONCLUSION

In this work, we establish a novel **anchor–multiplier factorization** paradigm which effectively decouples metric scale and relative geometry by leveraging sparse point anchors for metric guidance and a Gaussian-splatted multiplier for structural refinement. The proposed GSD framework demonstrates how 3D Gaussian Splatting can be utilized as a powerful and versatile **depth regulator**. A supporting theoretical analysis further confirms that the proposed factorization promotes training stability and facilitates easier convergence. Comprehensive experiments demonstrate the viability of our approach compared to state-of-the-art baselines across multiple benchmarks, underscoring the potential of 3DGS as a effective representation for scene-related 2D dense prediction tasks.

## Ethics Statement

We acknowledge our adherence to the ICLR Code of Ethics and address the following points:

**Datasets.** Our research utilizes only publicly available benchmark datasets that are widely adopted in the computer vision community. We conducted our experiments in full compliance with the intended use cases and licensing terms of these datasets. No personally identifiable information was involved or used in this study.

**Potential Impacts.** While our work is primarily fundamental research aimed at improving the accuracy and universality of depth estimation, we acknowledge that any perceptual technology carries a dual-use potential. The developed technique could, like other computer vision technologies, be applied in systems requiring geometric understanding, such as autonomous driving, robotics, and augmented reality. We strongly advocate for the responsible development and deployment of such technologies, with careful consideration of safety, fairness, and transparency. We are not aware of any immediate, specific societal harms that would arise solely from the methodological contributions of this paper.

**Research Integrity.** This work does not involve human subjects and therefore did not require IRB approval. We have strived for the highest standards of research integrity through transparent methodology, comprehensive experimental evaluation, and a detailed reproducibility statement to facilitate verification of our results.

## Reproducibility Statement

To facilitate the reproducibility of our work, we have made the following efforts:

**Theoretical Results.** All theoretical assumptions, theorems, and other claims are substantiated with explanations or proofs. Section Section 3 of the main text presents the core theoretical insights and provides a high-level overview of our analysis. For complete mathematical derivation, detailed step-by-step proofs for all assumptions and theorems are available in Section A.2.

**Experimental Setup and Methodology.** Our proposed GSD framework is described in detail in Section 4. The implementation details, including network architectures, hyperparameters, and training configurations for all experiments, are thoroughly documented in Section A.3.

**Data and Processing.** The datasets used in our experiments are publicly available and provided in Section A.4.

We are committed to supporting the research community and believe these resources will enable the replication of our results.

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

# A    APPENDIX

## A.1    DETAILED LITERATURE REVIEW

**Monocular Depth Estimation.**    Recently, monocular depth estimation has advanced rapidly towards foundation-scale models. Metric3D (Yin et al., 2023; Hu et al., 2024) proposes a canonical camera space transformation to solve the depth ambiguity caused by various focal lengths. Unidepth (Piccinelli et al., 2024) introduces pseudo-spherical output space representation to disentangle camera and depth representations, and a self-prompting camera module to support camera-free inference. Depth Anything (Yang et al., 2024a) and its successor Depth Anything V2 (Yang et al., 2024b) establish foundation models for monocular depth estimation through large-scale pretraining. Depth Pro (Bochkovskii et al., 2024) synthesizes high-resolution depth predictions based on a multi-scale vision transformer and an edge gradient loss. Marigold (Ke et al., 2024) presents a fine-tuning protocol for Stable Diffusion and a model for affine-invariant depth estimation. Geowizard (Fu et al., 2024) also distills the rich knowledge in the pre-trained Stable Diffusion. It proposes a geometry switcher that jointly produces depth and normal using a single model. DepthFM (Gui et al., 2025) presents a flow matching approach that improves sampling, data fidelity, training, and data efficiency. VGGT (Wang et al., 2025a) introduces a feed-forward neural network that can directly estimate all key 3D scene properties, including depth estimation. MoGe (Wang et al., 2025b) introduces an affine-invariant point map representation, an efficient point map alignment solver, and a multi-scale geometry loss for accurate monocular geometry estimation of open-domain images. Map Anything (Keetha et al., 2025) unifies local estimates into a global metric frame by using a factored representation of multi-view geometry (depth maps, ray maps, poses, and a global scale factor).

**Monocular Depth Completion.**    Depth completion aims to predict a dense depth map from an RGB image guided by a sparse depth map. NLSPN (Park et al., 2020) proposes a non-local spatial propagation module with confidence-aware affinity normalization to enhance relevant interactions and mitigate errors in depth propagation. SpAgNet (Conti et al., 2023) injects sparse depth points into a Scale-and-Place module instead of convolutions to handle uneven and sparse input distributions more robustly. CompletionFormer (Zhang et al., 2023) proposes a Joint Convolution-Attention and Transformer block that integrates local connectivity with global context. VPP4DC (Bartolomei et al., 2024) leverages the generalization capability of modern stereo networks to address depth completion by processing fictitious stereo pairs generated through a virtual pattern projection paradigm. BP-Net (Tang et al., 2024) propagates depth at the earliest stage to avoid directly convolving on sparse data. OMNI-DC (Zuo et al., 2024) introduces a multi-resolution depth integrator to handle extremely sparse inputs and employs a Laplacian loss to better model training ambiguity. Marigold-DC (Viola et al., 2024) builds on a pretrained latent diffusion model and injects the depth observations as test-time guidance via an optimization scheme that runs in tandem with the iterative inference of denoising diffusion. PromptDA (Lin et al., 2025) utilizes a low-cost LiDAR as a prompt to guide the Depth Anything model, enabling accurate metric depth output with resolutions of up to 4K. Prior Depth Anything (Wang et al., 2025d) introduces a coarse-to-fine pipeline that integrates precise but incomplete metric depth with complete but relative geometric predictions.

**Depth Estimation with 3DGS.**    3D Gaussian Splatting (3DGS) represents a cutting-edge paradigm in 3D reconstruction, where contemporary approaches increasingly exploit monocular depth estimation to enhance reconstruction fidelity and geometry. DepthSplat (Xu et al., 2025) leverages pre-trained monocular depth features for high-quality 3D Gaussian splatting and demonstrates its use as an unsupervised pre-training objective for depth models. CDGS (Zhang et al., 2025) leverages multi-cue confidence maps from monocular depth and sparse Structure-from-Motion depth to adjust supervision, thereby enhancing adaptive 3D Gaussian splatting. Mode-GS (Lee et al., 2024) integrates pixel-aligned anchors from monocular depth and generates Gaussian splats around them via residual-form Gaussian decoders. DHGS (Deng et al., 2025) combines 3D Gaussian splatting with depth-supervised learning using homogeneous coordinate embedding and adaptive monocular-SfM depth fusion, resolving scale ambiguity in distant views and enhancing local geometry via confidence-aware loss weighting. RDG-GS (Zhan et al., 2025) utilizes relative depth guidance to refine the Gaussian field, steering it towards view-consistent spatial geometric representations. CODN-GS (Hu et al., 2025) employs a normal-depth-normal transformation for accurate geometric feature capture and uses robust monocular depth supervision refined through global and local adjustments. 3DGS-Enhancer (Liu et al., 2024a) leverages 2D video diffusion priors to tackle 3D

view consistency by enforcing temporal consistency within video generation. Chung et al. (Chung et al., 2024) utilize an adjusted depth map from a pre-trained monocular model, which is aligned with sparse Structure-from-Motion points as a geometric reference. NexusGS (Zheng et al., 2025) leverages optical flow and camera poses to generate accurate depth maps, ensuring dense point cloud coverage and stable 3DGS training under sparse views. MoDGS (Qingming et al., 2025) introduces a 3D-aware initialization for learning deformation fields and employs a robust depth loss to guide the learning of dynamic scene geometry.

## A.2  PROOF OF VRT THEOREM

The basic notation is defined in Section 3.2. Now, we aim to characterize the formulation $\frac{\text{VAR}[g_\theta^{\text{new}}]}{\text{VAR}[g_\theta^{\text{trad}}]}$ step by step. For clarity, we begin with a single image sample for derivation. Based on Equation (2), the traditional paradigm gradient $g_\theta^{\text{trad}} = \frac{\partial \mathcal{L}}{\partial \theta}$ is determined by $\frac{\partial \mathcal{L}}{\partial f} \cdot \frac{\partial f}{\partial \theta}$. Hence, if we define the expectation over the gradient for each image sample as $\mathbb{E}[\frac{\partial \mathcal{L}}{\partial \theta}]$, then the variance $\text{VAR}[g_\theta^{\text{trad}}]$ can be represented by

$$\text{VAR}[g_\theta^{\text{trad}}] = \text{VAR}\left[\left(\frac{\partial \mathcal{L}}{\partial f} \cdot \frac{\partial f}{\partial \theta}\right)\right] \tag{25}$$

$$= \mathbb{E}\left[\left(\frac{\partial \mathcal{L}}{\partial f} \cdot \frac{\partial f}{\partial \theta}\right)^2\right] - \mathbb{E}\left[\left(\frac{\partial \mathcal{L}}{\partial f} \cdot \frac{\partial f}{\partial \theta}\right)\right]^2. \tag{26}$$

Under the Weak Dependence Assumption (Assumption 3), we have $|\text{Corr}(\frac{\partial \mathcal{L}}{\partial f}, \frac{\partial f}{\partial \theta})| \to 0$, indicating that the two terms can be regarded as nearly independent, and thus

$$\mathbb{E}\left[\left(\frac{\partial \mathcal{L}}{\partial f} \cdot \frac{\partial f}{\partial \theta}\right)\right] \approx \mathbb{E}\left[\frac{\partial \mathcal{L}}{\partial f}\right] \cdot \mathbb{E}\left[\frac{\partial f}{\partial \theta}\right]. \tag{27}$$

Since the model weights $\theta$ are usually randomly initialized at the beginning of training, the positive and negative gradients of $\frac{\partial f}{\partial \theta}$ are approximately symmetric around 0. Therefore, we can believe $\mathbb{E}\left[\frac{\partial f}{\partial \theta}\right] \approx 0$, indicating that the model sensitivity is approaching zero at the initialization period. And thus, we get $\mathbb{E}\left[\left(\frac{\partial \mathcal{L}}{\partial f} \cdot \frac{\partial f}{\partial \theta}\right)\right]^2 \approx 0$, and

$$\text{VAR}[g_\theta^{\text{trad}}] \approx \mathbb{E}\left[\left(\frac{\partial \mathcal{L}}{\partial f} \cdot \frac{\partial f}{\partial \theta}\right)^2\right] \tag{28}$$

$$= \mathbb{E}\left[\left(\frac{\partial \mathcal{L}}{\partial f}\right)^2 \cdot \left(\frac{\partial f}{\partial \theta}\right)^2\right] \tag{29}$$

$$\approx \mathbb{E}\left[\left(\frac{\partial \mathcal{L}}{\partial f}\right)^2\right] \cdot \mathbb{E}\left[\left(\frac{\partial f}{\partial \theta}\right)^2\right], \tag{30}$$

because $|\text{Corr}(\frac{\partial \mathcal{L}}{\partial f}, \frac{\partial f}{\partial \theta})| \to 0$, we can believe that $\frac{\partial \mathcal{L}}{\partial f}$ and $\frac{\partial f}{\partial \theta}$ are almost independent, as well as the squares of them.

Following the same property, we can derive a similar formulation of $\text{VAR}[g_\theta^{\text{new}}]$ such that

$$\text{VAR}[g_\theta^{\text{new}}] \approx \mathbb{E}[\mathcal{I}(S)^2] \cdot \mathbb{E}\left[\left(\frac{\partial \mathcal{L}}{\partial f}\right)^2\right] \cdot \mathbb{E}\left[\left(\frac{\partial \alpha}{\partial \theta}\right)^2\right], \tag{31}$$

and thus

$$\frac{\text{VAR}[g_\theta^{\text{new}}]}{\text{VAR}[g_\theta^{\text{trad}}]} \approx \mathbb{E}[\mathcal{I}(S)^2] \cdot \frac{\mathbb{E}\left[\left(\frac{\partial \alpha}{\partial \theta}\right)^2\right]}{\mathbb{E}\left[\left(\frac{\partial f}{\partial \theta}\right)^2\right]}. \tag{32}$$

Finally, under the Bounded Multiplier Assumption (Assumption 1) and the Metric-Depth Variation Assumption (Assumption 2), we can derive an approximated upper boundary over the whole dataset samples as

$$\mathbb{E}\left[\frac{\text{Var}[g_\theta^{\text{new}}]}{\text{Var}[g_\theta^{\text{trad}}]}\right] \lesssim \mathbb{E}[\mathcal{I}(S)^2] \cdot \frac{\kappa^2}{\Lambda^2}. \tag{33}$$

### A.3 IMPLEMENTATION DETAILS

**Metric multiplier transformation.** In practice, according to statistical counting (see the third sub-figure of Figure 2), we find that most of the $\alpha_{\text{gt}}$ values concentrate around 1. Moreover, given input resolution as $518 \times 686$ with only 86 anchor points (fairly sparse), $\alpha_{\text{gt}}$ mostly (within $3 \cdot \text{std}(\alpha_{\text{gt}})$) falls into the interval of $[0, 2]$, and thus we set the hyperparameter $\alpha_{\max} = 2$ and $\phi(\alpha_{\text{res}}^\theta) = 2 \cdot \alpha_{\text{res}}^\theta$ over most datasets to avoid multiple setting across different datasets.

**Stride for sparse anchors.** Following PromptDA, we introduce a sparse anchor interpolation method for synthetic datasets as mentioned above Equation (8). To align with PromptDA (Lin et al., 2025), we also downsample the GT depth map to low resolution ($192 \times 256$) and sample points with a stride of 7 under the first configuration, which has about 1,000 anchors per image. Since PromptDA and Marigold-DC have not released their training or sampling code, we reproduce the sampling method of PromptDA, simulating the noise of real LiDAR data, and expand it to more sparse settings. We also conduct the experiments for more sparse anchors with higher resolution($518 \times 686$), with each being a stride of 16(about 1,000 anchors), a stride of 32(about 300 anchors) and a stride of 64(about 80 anchors).

Notably, the number/percentage of depth anchors has a direct and significant impact on the results. For DepthLab (Liu et al., 2024b), it selected combinations of strokes, circles, and squares with **0.5%** to **1%** of pixels in the depth map, while Marigold-DC sampled **150** to **1,500** points at a resolution of 640×480. Our GSD can outperform with even fewer anchors. The sparsity of our sampling method can be found in Table 6.

Table 6: Stride, points and pct relationship

| stride | resolution | #points | percentage |
|---|---|---|---|
| 7 | $192 \times 256$ | 1000 | 2.0% |
| 16 | $480 \times 640$ | 1200 | 0.39% |
| 32 | $480 \times 640$ | 300 | 0.098% |
| 64 | $480 \times 640$ | 75 | 0.0024% |

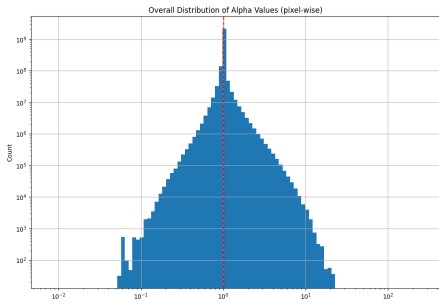

Figure 6: $\alpha$ distribution over pixels

**Training configurations.** Our model was trained on 8 NVIDIA V100 GPUs with 32GB of memory for 30 epochs, using a batch size of 2 and the AdamW optimizer with a learning rate of 2e-4. For Hypersim (Roberts et al., 2021) the image resolution is $518 \times 686$ downscaling to $192 \times 256$ and a LiDAR stride of 7 following PromptDA. For KITTI DC dataset we trained 10 epochs with the resolution of $378 \times 1252$. For VKITTI we trained 10 epochs with the resolution $364 \times 1204$ downscaling to $264 \times 912$ and a LiDAR stride of 7.

**Evaluation protocol and further discussions.** We compare our method to various baselines as shown in Table 1. All the methods have been trained on Hypersim, and we have fine-tuned them on KITTI Completion dataset for a fair comparison. The post-fusion method of Depth Anything v2 refers to scale and shift-based least squares alignment with relative depth prediction. We follow the official codebase to fine-tune Depth Anything V2 for metric depth estimation on their provided checkpoints for indoor and outdoor scenes, respectively, as DAv2 Metric. According to Prompting Depth Anything, their released checkpoint is pretrained on Hypersim and then the other two datasets. We reproduce the training process of PromptDA according to its paper, which achieves a much better result. Marigold-DC is a test-time training method that optimizes each dense depth map for several steps. As it is time-consuming, we only select one out of ten samples while taking the standard 50 inference steps for optimal performance.

Since most methods have not released their sampling sparse depth code for depth estimation datasets, it is a challenging task to align the anchor setting and re-evaluate all the metrics. We only conduct and re-evaluate four methods under our sampling and interpolation paradigm.

**Model details.** We inherit the pre-trained visual encoder DINOv2 from Depth Anything v2, using Vision Transformer Large, and take the last feature layer as a 1024-dimensional vector, denoted as $F_{vit}$. The CNN encoder has a downscale factor of 4, and the output $F_{cnn}$ dimension is 128. The channels of our fusion UNet are 128 for ViT-Large. Our Gaussian head decodes the refined features into 37 channels in detail: 1 for opacity, 2 for the origin offset of the ray, 3 for sphere scale (variance), 4 for quaternions related to the orientation/covariance, and 27 for SH coefficients (SH degree is 2). The total number of parameters in our model is 308 million (3.8 million trainable) for ViT-L and 25 million (3.1 million trainable) for ViT-S. Our model architecture is highly flexible, allowing for the replacement of the ViT with state-of-the-art DINOv3 or SAM modules. The CNN encoder can be replaced by a pre-trained ResNet, and the ViT encoder can be unfrozen for fine-tuning. We can even introduce another branch of image feature from CLIP or SigLIP2 for semantic enhancement. Equipped with the prior injection of the image encoding module pretrained on large-scale real-world datasets, we expect to achieve better experimental results.

## A.4 DATASET DETAILS

**Training dataset.** Our checkpoint is trained only on Hypersim (Roberts et al., 2021) for visualization and is compared with other methods in 1 and 2. It is worth noting that the Hypersim training set is an indoor synthetic dataset with only 59k training samples. We also train GSD from scratch on the KITTI Completion (Geiger et al., 2013) dataset—a real-world driving scene dataset with paired RGB images and sparse LiDAR depth for comparison, as shown in 1. Its semi-dense ground truth is derived from the temporal accumulation of consecutive LiDAR frames. For the outdoor zero-shot setting, we trained GSD on the Virtual KITTI dataset (Cabon et al., 2020) with about 21k samples.

**Evaluation datasets** In alignment with Prompting Depth Anything, we first make a comparison on the setting that downscales the ground-truth to the resolution of $192 \times 256$ with a stride of 7. Considering the potential application to high-resolution and authentic images, we also evaluated several stride settings on the Hypersim validation set. For outdoor scenes with irregular LiDAR anchors, we utilize the full validation split of 6,694 samples for the KITTI DC dataset (Cabon et al., 2020).

We also evaluated our GSD model in a zero-shot manner on five unseen real-world datasets. The evaluation datasets encompass both indoor and outdoor scenarios, covering a diverse range of image resolutions, sparse depth densities, acquisition devices, and noise levels. For NYUv2 (Silberman et al., 2012) and ScanNet Scannet(Dai et al., 2017), we evaluate at a resolution of $640 \times 480$. For KITTI (Cabon et al., 2020), we use a resolution of $352 \times 1216$, ETH3D Schops et al. (2017) uses $756 \times 1134$, and DIODE (Vasiljevic et al., 2019) uses $768 \times 1024$. All of the datasets above, except for KITTI DC, do not have LiDAR anchors; therefore, we sample the ground-truth with a stride of 16 for evaluation. KITTI Completion dataset is different from the KITTI dataset with sparse anchors and relatively denser depth ground truth. The zero-shot performance in 2 was trained on Hypersim from scratch.

972
973
974
975
976
977
978
979
980
981
982
983
984
985
986
987
988
989
990
991
992
993
994
995
996
997
998
999
1000
1001
1002
1003
1004
1005
1006
1007
1008
1009
1010
1011
1012
1013
1014
1015
1016
1017
1018
1019
1020
1021
1022
1023
1024
1025

## A.5 MORE VISUALIZATION RESULTS

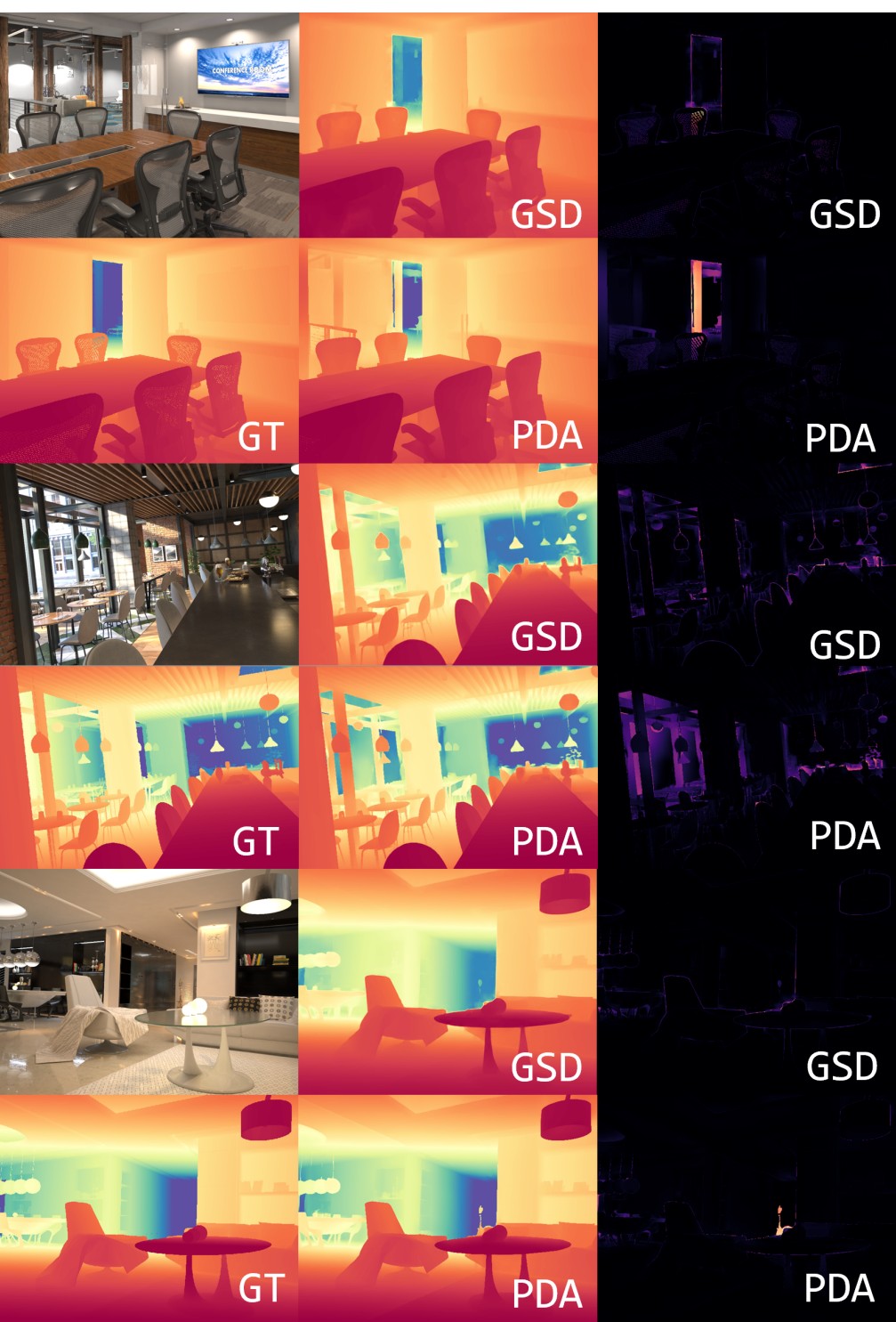

Figure 7: More visualization results compared with PromptDA. The first line of each scene covers RGB, GSD prediction and errors; the second line covers GT, PromptDA prediction and errors; stride=7, $d_{init}$ 192×256. Due to the opacity attribution of 3DGS, our GSD can better recognize transparent objects as shown in the above two images than PromptDA.

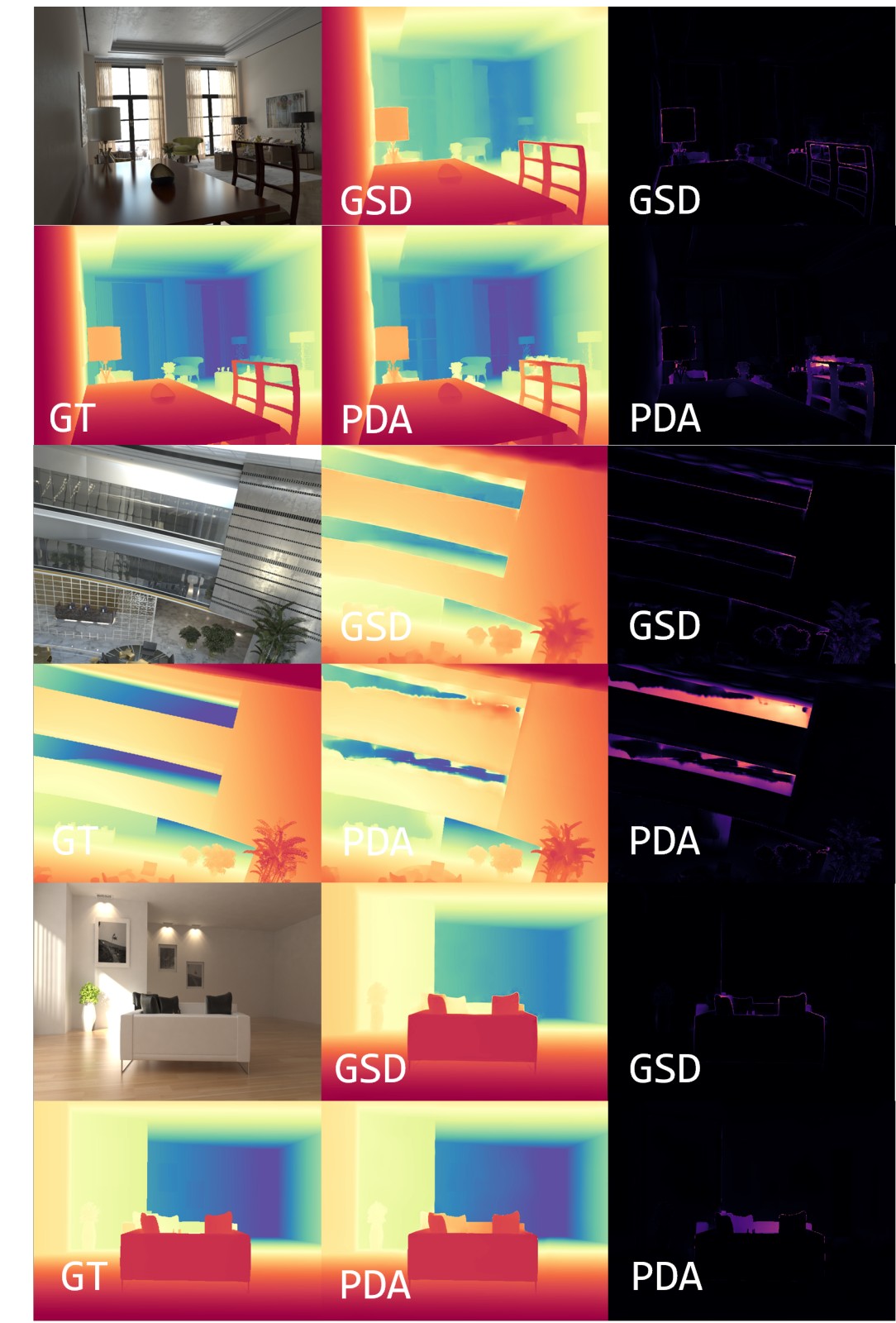

Figure 8: More visualization results compared with PromptDA. The first line of each scene covers RGB, GSD prediction and errors; the second line covers GT, PromptDA prediction and errors;stride=7, $d_{init}$ 192×256. Due to our strong regulation of depth anchors, GSD can follow the lidar prompt better, whereas PromptDA attempts to judge the distance by intuition such as the table behind the chair and the farther side of sofa.

### A.6 LLM USAGE STATEMENT

We acknowledge the use of large language models (LLMs) to assist in polishing the writing and improving the grammatical fluency of this manuscript. The human authors performed all ideation, technical development, experimental analysis, and final editing.

