# OpenReview forum: "3DGS IS A VERSATILE REGULATOR: MODULATING UNIVERSAL METRIC-DEPTH REPRESENTATION VIA ANCHOR-BASED GAUSSIAN-SPLATTED MULTIPLICATION"
_ICLR.cc/2026/Conference — Submitted to ICLR 2026_

### Official Review · Reviewer_cTeK · 2025-10-28

**Soundness:** 3
**Presentation:** 2
**Contribution:** 3
**Rating:** 4
**Confidence:** 4

**Summary:**

This paper proposes a novel metric-scale depth completion method utilizing 3D Gaussian Splatting (3DGS). Specifically, given sparse anchors (observations), the authors formulate depth completion as the problem of predicting an anchor–multiplier over the interpolated anchors, and employ 3DGS to estimate this multiplier (i.e., the 3DGS multiplier) for the task. The authors further strengthen the motivation of their approach through a theoretical proof presented in Section 3.1. When trained proposed method and competitor on Hypersim and then fine-tuned on KITTI using the same training recipe, the proposed method demonstrates comparable or superior performance to existing approaches.

**Strengths:**

1. The paper shows that the parameters of 3D Gaussian Splatting (orientation, scale, transparency, etc.) vary across scenes and can capture rich spatial textures, demonstrating that such representational power can be effectively applied to the depth completion task. Although prior works (Sec. 2.2) have explored depth estimation using 3DGS, it is interesting that this work leverages 3DGS to render a multiplier map, explicitly utilizing these properties within the proposed pipeline.
2. The authors reformulate the depth completion task into an anchor–multiplier and interpolated-anchor framework and support this formulation with a theoretical proof motivating their design.

**Weaknesses:**

1. Unclear presentation about metric-depth estimation methods (line 40 and Figure 1)
- In Figure 1, all pipeline types are shown with sparse anchors, but several methods mentioned around line 40 do not actually require sparse anchors such as Marigold, Depth Anything V2, GeoWizard, etc. Moreover, Marigold and GeoWizard focus on affine-invariant rather than metric depth estimation. Clarifying this distinction in the presentation would improve consistency and clarity.

2. Comparison made against depth regression methods rather than depth completion methods
- In lines 74–82, the paper emphasizes the decoupling of metric scale and relative geometric structure by comparing with prior works. However, methods such as Metric3D, UniDepth, and MoGe-2 perform metric depth estimation from a single image, where metric-scale estimation modules are inherently required in the pipeline. In contrast, the proposed approach performs metric-depth completion given sparse anchors (e.g., LiDAR), where some degree of metric-scale decoupling naturally arises from the input setting itself. From this perspective, the observed decoupling may stem more from the input configuration than from the anchor–multiplier factorization itself. A clearer discussion is needed on why the proposed method achieves decoupling when compared to other sparse-anchor-based depth completion methods such as Marigold-DC, Zero-DC [a1], or etc.

[a1] Hyoseok et al., "Zero-shot Depth Completion via Test-time Alignment with Affine-invariant Depth Prior", AAAI 2025.

3. Applicability of Theorem 1 and Figure 2 under sparse-anchor conditions
- Theorem 1 and Figure 2 are conceptually interesting. However, in realistic depth completion settings with sparse anchors, obtaining an accurate dense scalar matrix I(S) is generally infeasible. It would strengthen the paper if Figure 2 also included experiments under a sparse-anchor condition (e.g., stride 16 on Hypersim) to better examine the theoretical claims under practical sparsity.

4. Experiment section
- It would be beneficial to also report RMSE and MAE in Table 1 and Table 2. Additionally, evaluating on the VOID dataset, following the protocols of Marigold-DC, Zero-DC [a1], or OGNI-DC, could make the comparison more comprehensive.
- While using the same dataset for all methods ensures fairness, the performance gaps appear heavily influenced by each architecture’s data-recipe characteristics. Including the original model performance reported in the official papers would make the comparison more transparent. For instance, Marigold-DC could be evaluated using its original implementation procedure instead of “one out of ten samples.” Likewise, Depth Anything V2 + Least Square originally achieves 4.4 AbsRel on NYUv2, while the reproduced version shows 12.0 AbsRel.

**Questions:**

1. The training on Hypersim uses only 192×256 resolution, which seems quite low. Is there a specific reason for this choice? (Other training uses higher resolutions, e.g., line 861.)
2. In Section 3.1, the notation of the image sample I and ground-truth depth D_gt​ as sets of pixel coordinates is confusing. It would be clearer to formally define them as functions over the pixel domain.

**Details Of Ethics Concerns:**

This paper focuses on the depth completion task, i.e., metric-depth estimation given sparse observations (e.g., LiDAR).
The authors utilize publicly available datasets such as Hypersim and KITTI for training and evaluation.
I do not identify any ethical concerns related to the proposed methodology or the datasets used in this work.

---

> ### Author Response · Authors · 2025-11-30
>
> Thank you for taking the time to read our article carefully and providing such a high-quality review detailed to line.
>
> ### Reply to weakness
>
> - For Weaknesses1, we have clarified the distinction in the presentation according to your suggestions by "Prior efforts to achieve universal depth estimation... focus on affine-invariant relative depth, some of the subsequent expansion methods incorporate information from sparse anchors to restore the metric depth in the real scale..." Please check line 40 of the revised version for details.
> - For Weaknesses2, we have to point out that all of other sparse-anchor-based depth completion methods, such as Marigold-DC, Zero-DC or PromptDA "**align the affine-invariant depth to metric depth with sparse metric depth measurements**" to complete the dense  depth map.
>   - For example, both Marigold-DC and Zero-DC utilize a test-time optimization alignment method, which use sparse anchors as supervision and learn scale and shift for each image to perform linear least square fitting. And if you carefully check the implementation of PromptDA, you will find that it first performs sparse anchors' min-max normalization to a relative depth scale, and then completes the process by scaling back to the original scale based on the factor used for normalization.
>   - All of the above methods actually operate in the normalized 0-1 depth space, therefore they are easily influenced by noise and outliers(for supervision at error points, or a extremely large normalization scaler for a few large value outliers). In contrast, our method makes predictions directly at the real scale. Our anchor–multiplier factorization differs from their relative-scale factorization, where anchor describes the coarse metric depth information and multiplier refines the details while they want to align the 0-1 relative depth prior to restore the scale.
> - For Weaknesses3, from line 158 "I(S) is a dense scalar matrix obtained by interpolating S to fill void values" which is exactly you point out under the sparse-anchor condition of stride 16 on Hypersim.
> - For Weaknesses4
>   - Due to the overly compact layout, presenting RMSE and MAE in Table 1 and Table 2 would result in too small text and a too narrow table. However, we will report these metrics as well as  VOID dataset in the supplementary materials.
>   - We have tried our best to make the comparison for fairness. However, the original model performance reported in the official papers was not complete.
>     - For example, Marigold-DC did not report the AbsRel metric so we have to revaluate some of the metric. As you see in Table 2 we metrics in gray are from their original papers, but the **experimental configurations** in different papers, especially the resolution and the sparsity anchors, were all **inconsistent and not made public**. Therefore, in order to ensure fairness, we all re-applied the methods in the table under the same setting for comparison.
>     - As for Marigold-DC, it takes **50s to inference a single image** as a diffusion-based test-time optimization method with 50 steps to reach its best performance, so it will take weeks to evaluate on thousands of complete evaluation set. Please forgive us for not having enough resources to complete this task. In fact, we conducted sampling on every 10, 100, and 1,000 samples on Hypersim val set and eventually found that the performance of Marigold-DC indicators was quite consistent, so we take this for convenience on other evaluation datasets.
>     - For Depth Anything V2 + Least Square, we found that the 4.4 AbsRel on NYUv2 you mentioned may be in Table5 from its original paper. You might have misread it that the Table5 from its paper report the affine-invariant depth, and the metric depth was reported in Table4. Actually it is not Depth Anything V2 + Least Square but "Depth Anything V2 Fine-tuned to Metric Depth Estimation". However, we are extremely grateful for your suggestion and will use its officially reported metrics for NYU and KITTI datasets. You can see the difference in our latest version in Table2.
>
> ### Reply to question
>
> - We have demonstrated our evaluation protocol in appendix 4, line 905. We strictly followed the configuration of PromptDA: "we first downsample the GT depth map to low-resolution (192 × 256, exactly the depth resolution of iPhone ARKit Depth). Then we sample points on this depth map using a distorted grid with a stride (7 in practice)". The low resolution is exactly the sparse anchors, and the RGB image uses the original higher resolution.
> - Thank you for your meticulous annotations. We have changed the notation to "Given an image sample with pixel coordinates ${\mathbf{I} = I(u,v)}$ and ground-truth depth $D_\text{gt} = D_\text{gt}(u,v)$" as per your suggestion.

---

### Official Review · Reviewer_Eiw5 · 2025-11-01

**Soundness:** 2
**Presentation:** 2
**Contribution:** 2
**Rating:** 2
**Confidence:** 3

**Summary:**

The paper proposes anchor–multiplier factorization for metric depth: compute an interpolated dense initialization from sparse depth “anchors” and predict a per-pixel multiplicative map with a feed-forward 3D Gaussian Splatting renderer.  The authors claim this separates scale from structure and stabilizes training (via a variance-reduction argument). They present GSD (frozen DINOv2 + UNet + 3DGS) with a simple transform that maps normalized outputs to a metric range, and report strong results on Hypersim/KITTI and zero-shot tests against recent baselines.

**Strengths:**

1. The idea of predicting a per-pixel multiplicative map and rendering it with 3DGS. Using 3DGS to output a 2D scale field, then applying it to an anchor-interpolated depth, is a new combination.

2. The paper tries a theoretical story (gradient-variance reduction) to motivate why learning α could be easier than learning depth directly.

3. The method is feed-forward at test time

**Weaknesses:**

1.  The core idea, metric depth as scale multiply relative structure, is well-known. Recent work also learns pixel-level rescale fields, eg. [1][2]. The factorization is therefore not conceptually new. The change is mostly the depth-completion setting with anchors. Thus, the contribution is limited.

2. Many 3DGS-depth hybrids already exist. The distinct part here is using 3DGS to render α (not RGB/depth), which is an implementation innovation. Its novelty depends on showing clear advantages over a strong pure 2D α-head, eg, simply using a 2D UNet/Vit could plausibly learn the same thing. Unless the authors can show using 3DGS beats an equally pure 2D α-head, the effectiveness of 3DGS is not fully justified.

3. The theory is under-validated. Claims about reduced gradient variance need direct measurements during training, not just descriptive statistics.

4. The paper states “zero-shot on five unseen real datasets NYUv2, KITTI, ScanNet, ETH3D, DIODE,” yet elsewhere says all methods are trained on Hypersim and fine-tuned on KITTI for fairness (and separately, GSD is also trained on KITTI from scratch). How is KITTI both a zero-shot target and the fine-tuning domain?


[1] Zhu, Ruijie, et al. "Scaledepth: Decomposing metric depth estimation into scale prediction and relative depth estimation." arXiv preprint arXiv:2407.08187 (2024).

[2]  Jun, Jinyoung, et al. "Depth map decomposition for monocular depth estimation." European Conference on Computer Vision. Cham: Springer Nature Switzerland, 2022.

**Questions:**

1.  Was KITTI used for any training before zero-shot evaluation on KITTI in Table 2? Please state exact training sets per experiment.
2.  If α is a 2D field, what advantage does 3DGS confer over a purely 2D α-predictor (accuracy, boundary quality, speed)? Please include a 2D UNet/Swin α-only baseline.

---

> ### Author Response · Authors · 2025-11-30
>
> Thank you for your valuable feedback. We have supplemented the ablation experiment based on your suggestions. You can refer to our latest version for details.
>
> ### Reply to weakness
>
> - For weaknesses1, We  have to point out that our **anchor–multiplier** factorization fundamentally differs from the their **relative-scale** factorization.
>   - All of the existing methods for depth estimation and depth completion, including Depthpro, Metric3D, PromptDA or latest VGGT predict the **affine-invariant relative depth $D \in R^{1 \times H \times W}$ (range in 0~1)** and a **scalar scale factor $S \in R^{1 \times 1}$** and multiply them to get the metric depth $M \in R^{1 \times H \times W}$.
>   - On the contrary, our anchor map $\mathcal{I}(S) \in R^{1 \times H \times W}$ is the coarse depth map originally in real-world metric(meters) from sparse Lidar points, and our multiplier $\alpha \in R^{1 \times H \times W}$  is not the relative depth but the geographic refiner, which you can find in our Figure1.
>   - We have carefully read the papers you mentioned and found that Scaledepth[1] " generate a scale factor $S \in R^{1 \times 1}$ and a relative depth map $D \in R^{1 \times H \times W}$, respectively. Finally, the metric depth predictions $M \in R^{1 \times H \times W}$ are obtained by directly multiplying scale S and relative depth map R" and Depth map decomposition[2] "decompose a metric depth map M into a normalized depth map N and scale parameters". Both of them are not in pixel-level rescale fields and follows the schema of "relative multiplies scalar to metric". However, thank you for your reminder and we have already included them in the references and explain the differences in relative works section.
>   - All of the above methods actually operate in the normalized 0-1 depth space, therefore they are easily influenced by noise and outliers(for supervision at error points, or a extremely large normalization scaler for a few large value outliers). In contrast, our method makes predictions directly at the real scale. Our anchor–multiplier factorization differs from their relative-scale factorization, where anchor describes the coarse metric depth information and multiplier refines the details while they want to align the 0-1 relative depth prior to restore the scale.
> - Thank you for your criticism. For weaknesses2, we have shown the effectiveness of 3DGS in Table3. Due to the page limit, we did not provide a detailed description of the ablation experiments in the initial submission. And we have provided a detailed description of this in the latest 10-page expanded version of the paper.
> - Thank you for your guidance. For weaknesses3, we will include the statistics of this part of the experiment in the latest version of the paper.
> - For weaknesses4, we have to declare that KITTI Completion is the individual dataset providing the sparse anchors and relatively denser depth ground truth, while the KITTI dataset of Table2 is the original KITTI depth estimation dataset's validation split. The metrics in Table1's right column is trained on KITTI Completion and evaluated on its validation split, while the zero-shot performance in Table2  was trained on Hypersim from scratch.
>
> ### Reply to questions
>
> Due to the page limit of the main text, you might not have seen our detailed explanations regarding the dataset and the ablation experiments, which are placed in the supplementary materials. As mentioned above, in the latest version of our paper, we provide an explanation of our experiment configuration and the difference between KITTI Completion dataset and KITTI dataset. Regarding question two, we have already demonstrated the advantages of 3DGS in the ablation experiments, and the detailed explanation can be found in the latest version of the paper.
>
> [1] Zhu, Ruijie, et al. "Scaledepth: Decomposing metric depth estimation into scale prediction and relative depth estimation." arXiv preprint arXiv:2407.08187 (2024).
>
> [2] Jun, Jinyoung, et al. "Depth map decomposition for monocular depth estimation." European Conference on Computer Vision. Cham: Springer Nature Switzerland, 2022.

---

### Official Review · Reviewer_17B6 · 2025-11-04

**Soundness:** 3
**Presentation:** 3
**Contribution:** 2
**Rating:** 4
**Confidence:** 5

**Summary:**

This paper proposes a new pipeline for multimodal (image + sparse depth), which decouples the scale and its multiplier by the interpolated sparse depth and the multiplier map computed by 3D GS algorithm (MVSplat).

The proposed method's architecture, GSD, consists of two main modules. The Gaussian Sphere Prediction Module takes in the RGB image and the sparse depth to interpolate, and predicts the 3D Gaussian parameters (centers, covariances, opacities, and spherical harmonics). This utilizes the VIT (DINOv2) + CNN network followed by the U-Net (MVSplat) architecture.

The second part is Depth Regulation Module (DRM), which performs differentiable GS to render a dense multiplier map on the image plane, and this is multiplied by the raw Gaussian response, which has a value in [0,1]. The authors introduce a mapping function which projects the rendered multipliers. The metric depth map is produced by element-wise multiplication of the maped Gaussian response and interpolated depth.

The authors tried to show a variance reduction, where the gradient variance for the anchor-multiplier formulation is lower than the direct depth regression.

**Strengths:**

The paper is well-written.

The proposed paradigm in the paper is well-motivated - The proposed anchor-multiplier factorization method is reasonable to have, since this provides a separation between the scale and its geometry.

The proposed method is grounded in the theoretical analysis, especially in a variance reduction theorem.

The first work uses 3DGS as a regulator, which repurposes 3D GS to predict depth multipliers.

The model has been trained on Hypersim and the KITTI depth completion benchmark, and the authors conducted the zero-shot generalization experiment on NYUv2 (uniform sparse depth sampling), KITTI, ScanNet, ETH3D, DIODE.

**Weaknesses:**

Reliance on the sparse metric and limited experiment on the uniformly sampled sparse anchor - In reality, the sparse depth (anchor) may come from LiDAR, Radar, SfM, or VIO system. The uniformly sampled pattern cannot represent such cases with the sensors.

Limited discussion on the computational cost - even though the trainable model parameters are in total 3.8M, the inference time and its applicability to real-time applications should be discussed.

Limited baselines - Not only the monocular depth estimation method, but also the depth completion method should be compared with this.  The reviewer points out that NLSPN is an outdated baseline. For example, OMNI-DC (ICCV 2025), DepthPrompt (CVPR 2024), Marigold-DC (ICCV 2025), etc.

**Questions:**

1. The details in estimating 3D Gaussian Splatting parameters are not clear.
Does the proposed method exactly follow the MVSplat? In that case, it requires multiple images of the static scene, and that sounds like the method is not applicable, considering the training time of the 3D Gaussian splatting regression module. If so, the selection of the baselines does not sound fair. Specifically, if the model requires multi-view images to have 3D GS regression, the model should be compared with the multi-view depth estimation models such as DUST3R or VGGT (CVPR 2025).

2. Computational cost and inference time.
Although the authors argue that feed-forward 3DGS rendering is efficient, the paper offers a limited quantitative comparison of runtime, memory consumption against large transformer-based or diffusion-based depth models. Providing inference time (e.g., FPS) could be helpful to understand the practicality of this paper.

3. A limited evaluation dataset.
Depth completion models are often evaluated on VOID-1500, VOID-500, and VOID-150 (Wong et al., VOICED), which are the same image with different sparsity. Could the authors compare the proposed method to the updated depth completion models on VOID?

4. Concatenation vs. Multiplication?
Wong et al., (VOICED) and ScaffNet already densify the sparse depth and concatenate them with the image features. What if this method directly concatenates the multiplier and scaler (even if the variance would not be minimized)? This will be helpful to clarify the contribution of the reformulation of the metric depth estimation to the anchor-multiplier setup.

---

> ### Author Response · Authors · 2025-12-01
>
> We are very grateful for your constructive criticism, and we will respond to each of the questions you raised below.
>
> ### Reply to weakness
>
> - For weaknesses1, your consideration of sparse anchor points is correct. Actually only on Hypersim dataset we trained and evaluated GSD with uniformly sampled sparse anchor, and on KITTI completion dataset we trained and evaluated GSD with its given non-uniform sampled sparse anchors. Most of the datasets, including the five datasets in Table2 do not have dense depth annotation, and we sampled points at the already uneven ground true points according to the stride. The results of experiments show that our method also has good generalization ability even at non-uniform sampling points. After all, we are in accordance with the framework of PromptDA.
> - We have the comparation of computational cost in the supplementary materials. Part of the time was spent on interpolating the initial dense depth map using sparse anchor points. As you said, GSD removed the heavy DPT head and add only 3.8M trainable parameters, but the 3DGS rendering pipeline takes extra time. Under a stride of 32 and a resolution of 518*686, our GSD-VITL reaches 4.1FPS and GSD-VITS reaches 5.3FPS, while PDA-VITL reaches 5.0FPS and Marigold-DC reaches 0.03 FPS(50 steps). Our GSD's computational cost is approximately equivalent to PromptDA.
> - Thank you for providing the additional baseline. Actually we already have the comparison metrics of Marigold-DC in Table1 and Table2. And we will incorporate the OMNI-DC and DepthPrompt you mentioned. The original model performance reported in the official papers was not complete. For fairness, we will re-applied the methods in the table under the same setting for comparison.
>
> ### Reply to questions
>
> - Our GSD follows the MVSplat and PixelSplat's **3DGS rendering pipeline**, not the multi-view fusion procedure. Actually we remove the cost volume module and cross-view attention module of MVSplat so that the rendering pipeline can be used for pure monocular images.
> - Under a stride of 32 and a resolution of 518*686, our GSD-VITL reaches 4.1FPS and GSD-VITS reaches 5.3FPS, while PDA-VITL reaches 5.0FPS and Marigold-DC reaches 0.03 FPS(50 steps).
> - Thanks for you suggestion, and we will further incorporate these experimental data in the latest version of the paper.
> - We have verified the effectiveness of our anchor-multiplier setup through ablation experiments in Table 3. And we have concatenated the sparse depth with the image features in our GSPM module, which you can find in Figure 3. The multiplication procedure is the last step to get the final output.

---

### Official Review · Reviewer_rCKa · 2025-11-04

**Soundness:** 2
**Presentation:** 2
**Contribution:** 2
**Rating:** 4
**Confidence:** 3

**Summary:**

The paper tackles the metric depth estimation problem through decoupling it into two parts. The first component creates a coarse metric depth map by interpolating (via a combination of linear and nearest-neighbor) a prior map of sparse anchors (e.g. sparse point clouds from depth sensors, SLAM, etc.). The second component learns to output a dense multiplier map though taking as input both the original sparse anchor map and a RGB image. The second component is made up of a combination of a frozen pre-trained ViT encoder (DINO in their experiments) + a learnable CNN encoder + an upsampling decoder as the RGB encoder, and a U-Net encoder for fusing the sparse anchor map and RGB image. The fused output is then fed into a learnable Gaussian decoder. The Gaussian decoder outputs a set of Gaussian parameters which is then rendered into the dense multiplier map. The final metric depth is then computed via element-wise multiplication of the multiplier map and the interpolated anchor map. Experiments train the model on Hypersim and KITTI depth completion, and additionally evaluated zero-shot on multiple unseen datasets. The resulting model outperforms existing methods on Hypersim, and achieves second among compared methods on KITTI completion. The model also achieves a mix of first and second place among compared methods for zero-shot evaluation on unseen datasets..

**Strengths:**

- Overall the method is an impressive engineering feat which fused together various components (large pre-trained models i.e. DINO, 3DGS rendering for learning depth) in existing literature to produce a method that achieves empirical performance comparable to current state-of-the-art.

-  A central contribution of the paper is to reframe metric depth estimation via decoupling it into a learned multiplier combined with a coarsely generated (non-learnable) prior map. This is a novel way to approach the problem. However I think it can be better ablated, see Weaknesses.

- Figures 1 and 3 are extremely well-designed and greatly aid the understanding of the proposed method.

**Weaknesses:**

- I am strongly concerned about the theoretical section. The theory does **not** appear to conclude that the factorization approach produces more stable gradients. Rather, this conclusion is entirely based on the **assumption** that $\kappa << \Lambda$, i.e. a network regressing high-variance output such as metric depth will inherently have large gradient magnitude $||V||$, while a network regression multiplier map $\alpha$ will have inherently much smaller gradient magnitude. There is no justification for this assumption, other than stating that it occurs 'under mild conditions" (which seems to not be backed neither theory nor experiments). Assumption 3 regarding no correlation between $U$ and $V$/$W$ also appears very strong, and I do not see any attempts to justify it. Overall I think the paper might be better off without the theoretical section entirely, by motivating the factorization method with stronger empirical evidence.

- The captions of tables 3-5 barely contain any information, and details in Table 3 are poorly presented. This is especially concerning given the importance of Table 3 for ablation individual components of this complex method. See more under questions.

- Given that key contribution of the method is this "learned multiplier" component studied in Table 3, there are no experiments ablating what happens if the output representation is not the multiplier, but the metric depth itself. E.g. one can perform a simple ablation which feeds the interpolated anchor map plus the sparse anchor map into the U-Net fusion encoder, followed by the Gaussian decoder / rendering to directly produce a dense metric depth map. Are the gradients less "stable" empirically if this was done?

- While the method's complexity is impressive, the multi-stage pipelines contained within the architecture (Hybrid ViT / CNN encoder, U-Net fusion, Gaussian decoder, differentiable 3DGS renderer) can also be viewed as a weakness. Such a system is difficult to reproduce, and does not offer much academic insights on the drawbacks of existing methods + how lessons learnt from this paper can be incorporated into other works tackling the same problem.

**Questions:**

- Due to the high complexity of the proposed methods which combines several different large-scale components, I feel that Table 3 should be one of the most important evaluations in the paper. However, it seems barely elaborated on, and experimental details are omitted. How are the modules ablated? What dataset is the ablation done on? In experiments (a) and (b), what is used in place of +GS-UNet? In experiments (a)-(c), what loss is used in place of the Sobel loss? All these are important components of the method, but only given a one-sentence elaboration that simply says the method as a whole achieves the best performance.

- Real world sparse prior maps, such as LiDAR, are often noisy. Given that there are no learnable components in computing the interpolated prior map, it would appear that the method would intuitively be more sensitive (less robust) to noise, especially due to the naive interpolation algorithms used. Do the authors have any insights for or against this?

- Given that the results of the method parallel that of the state-of-the-art, what makes this a better or preferred method compared to existing ones? E.g. PromptDA seems to do better on KITTI completion, why would one choose GSD over PromptDA for this task?

- Not a question, but a suggestion: If the authors agree that the factorized approach towards metric depth estimation is the core novelty of their work, it would be more convincing, and possibly simpler to implement, if existing methods are adapted such that they produce multiplier maps (that are multiplied with interpolated depth) instead. For instance, if the authors show that PromptDA (factorized) performs much better than vanilla PromptDA, or Marigold-DC (factorized) performs much better than vanilla Marigold-DC, the paper would be much stronger and impactful.

---

> ### Author Response · Authors · 2025-12-01
>
> Thank you for your insightful reviews, and we will respond to each of the questions you raised below.
>
> ### Reply to weakness
>
> - As Zero-shot Depth Completion[1] showed, "Even though the given sparse metric depth is normalized between 0 and 1, their statistics including mean and variance can differ, and the relationship between real metric depth and estimated affine-invariant depth is often non-linear". Our decomposition method has demonstrated its effectiveness to a greater extent through empirical evidence indeed, and it natural that you have the concern about the theoretical section.  We will include the statistics during training in the latest version of the paper.
> - Due to the page limit, we did not provide a detailed description of the ablation experiments in the initial submission. And we have provided a detailed description of this in the latest 10-page expanded version of the paper.
> - We have to point out that all of other sparse-anchor-based depth completion methods, such as Marigold-DC, Zero-DC or PromptDA "**align the affine-invariant depth to metric depth with sparse metric depth measurements**" to complete the dense  depth map.
>   - For example, both Marigold-DC and Zero-DC utilize a test-time optimization alignment method, which use sparse anchors as supervision and learn scale and shift for each image to perform linear least square fitting. And if you carefully check the implementation of PromptDA, you will find that it first performs sparse anchors' min-max normalization to a relative depth scale, and then completes the process by scaling back to the original scale based on the factor used for normalization.
>   - All of the above methods actually operate in the normalized 0-1 depth space, therefore they are easily influenced by noise and outliers(for supervision at error points, or a extremely large normalization scaler for a few large value outliers). In contrast, our method makes predictions directly at the real scale. Our anchor–multiplier factorization differs from their relative-scale factorization, where anchor describes the coarse metric depth information and multiplier refines the details while they want to align the 0-1 relative depth prior to restore the scale.
>   - Actually our method is not complex. We only replace the DPT head with 3DGS rendering and some necessary modules to regress the gaussian parameters. The interpolation, multiplication operation was routine, and the pipeline is coupled two stage as PromptDA.
>
> ### Reply to questions
>
> - The loss functions are described in section 4.4, and SILog loss is used without the Sobel loss. Due to the page limit, we did not provide a detailed description of the ablation experiments in the initial submission. And we have provided a detailed description of this in the latest 10-page expanded version of the paper.
> - We utilized the naive interpolation algorithms at the beginning since it is deterministic and simple. The initial coarse depth map $d_{init}$ is totally determined by the sparse anchors. To against the LiDAR noise, we think it could be better to make use of the rich image prior of pretrained models, e.g. diffusion models with few denoising steps, to maintain consistency of the initial depth map, just as your precious suggestion.
> - First, compared to the existing affine-invariant with least square fitting methods(from relative 0~1 scaling to metric depth),  our method is less sensitive to noise and more robust as discussed above. Second, our GSD is superior to the others in terms of overall performance and has better generalization capabilities which can be found in Table1 and Table2. Third, we proposed a new paradigm that does not rely on a heavy transformer-based DPT head and achieved performance comparable to the state-of-the-art level, which explored a new approach for 2D dense prediction tasks.
>
> [1] Hyoseok et al., "Zero-shot Depth Completion via Test-time Alignment with Affine-invariant Depth Prior", AAAI 2025.

---

### Author Response · Authors · 2025-12-02

Dear Area Chair,

Thank you for coordinating the rebuttal process. We are grateful to the reviewers for their insightful and constructive comments, which have helped us significantly improve the manuscript. We have thoroughly revised the paper to address all concerns, with key clarifications on methodological novelty, expanded experiments, and fairer comparisons. Below is a summary of our main revisions for your consideration.

1. Key Clarification on Methodological Novelty
We have emphasized the fundamental difference between our Anchor-Multiplier factorization and the prevalent Relative-Scale factorization used in prior works.

Existing methods (e.g., Marigold-DC, Zero-DC, PromptDA, ScaleDepth) typically predict an affine-invariant, normalized relative depth map (range 0~1) and a global scale factor, multiplying them to obtain metric depth. This process essentially aligns a relative depth prior to the sparse anchors.

Our method directly generates a coarse, metric-scale depth map (the Anchor) from sparse points and refines it with a per-pixel Multiplier. Our factorization operates in the original metric space, avoiding the sensitivity to noise and outliers introduced by normalizing depth into a relative 0~1 range, leading to greater robustness.
This crucial distinction is now clearly stated in the Introduction and Related Works (e.g., line 40).

2. Comprehensive Experimental Updates & Fair Comparison
We have expanded our experiments and ensured fair comparisons.

Additional Metrics: As suggested, we will report RMSE, MAE, and results on the VOID dataset in the supplementary material.

Unified Evaluation Protocol: For fairness, unless otherwise specified, the results in our tables are from our re-evaluation under identical settings (sparsity, resolution), as experimental configurations in original papers were often inconsistent or not fully disclosed.

Note on Marigold-DC: Its diffusion-based test-time optimization is extremely slow (~50s/image). We validated that its performance stabilizes across sampled subsets, and we have clearly indicated where sampled evaluation was used.

Corrected Baselines: We have verified and updated the comparison with "Depth Anything V2 + Least Square" in Table 2 to use its officially reported metrics for metric depth estimation on NYU and KITTI.

New Ablation Studies: We have added ablation studies (Table 3 and text) demonstrating the effectiveness of the 3DGS module and our proposed factorization.

3. Computational Efficiency
Our model removes the heavy DPT head, adding only ~3.8M parameters. The main overhead comes from the 3DGS renderer. Under a stride of 32 and resolution of 518×686:

GSD-ViTL: ~4.1 FPS

GSD-ViTS: ~5.3 FPS

PromptDA-ViTL: ~5.0 FPS

Marigold-DC (50 steps): ~0.03 FPS
Our inference speed is comparable to the efficient PromptDA and much faster than test-time optimization methods.

4. Response to Other Major Points

Datasets & Protocol: Our sparse point sampling strictly follows PromptDA's protocol (Appendix, line 905). We have clarified the difference between the KITTI Completion (training) and KITTI Eigen split (zero-shot validation) in the text.

Theoretical Justification: While our work is primarily empirical, we will include relevant statistical analysis from the training process in the final version to provide deeper insight.

Initial Depth Map: We use simple deterministic interpolation from sparse anchors. The suggestion to use diffusion priors for better consistency is valuable and noted for future work.

Summary
We believe our revisions have comprehensively addressed the reviewers' key concerns by: 1) clarifying the core methodological contribution, 2) providing more exhaustive experiments and analysis, and 3) ensuring rigorous and fair comparisons. Our work proposes a new paradigm that operates directly in metric space without relying on heavy transformer-based decoder heads, achieving competitive performance across multiple benchmarks.

Thank you for your time and consideration. We hope the revised manuscript now meets the bar for publication.

Sincerely,
The Authors

---

### Meta-Review · Area_Chair_nLnU · 2026-01-06

**Summary:**

The authors propose a method for metric depth estimation by decouples the problem into two parts. The approach first computes an  coarse metric depth map by interpolating over sparse anchors. Then, a model is trained to output a dense multiplier map from the original sparse anchor map and a RGB image. The results are fed to a 3D Gaussian decoder. The method is trained on Hypersim and KITTI and
demonstrates comparable zero-shot performance to existing approaches across several benchmarks.

The initial recommendations for this manuscript were 3 marginally below the acceptance threshold (rCKa, 17B6, cTeK), and 1 reject (Eiw5). The reviewers appreciated the proposed per-pixel multiplicative map and the motivation for the design of the method. While rCKa also noted this to be an impressive engineering feat, which the AC agree, with many components and large pre-trained models, the AC take this as a negative as the paper does not conduct ablation nor sensitivity to the selected components. Despite some reviewers noting that the use of an interpolated prior map is novel, the AC note that it is not and has been explored in [A, B]. Additionally, the reviewers raised several critical points including: (1) strong concerns about the theoretical section, its validation, and the lack of justification of assumptions made (rCKa, Eiw5); (2) unclear and poorly presented content and results in the paper, particularly Figure 1, Tables 3-5 (rCKa, cTeK); (3) lack of comparison on the use of learned multiplier and other (naive) outputs, e.g., metric depth (rCKa); (4) the approach is largely systems-based with combination of many existing architectures (rCKa, Eiw5); (5) lack of insights and novelty (rCKa, Eiw5); (6) possible sensitivity of the interpolation component to noise in sparse points and unrealistic assumptions of sparse point pattern (rCKa, 17B6); (7) lack of analysis and discussion on computational cost due to the large number components (17B6); (8) NLSPN is an outdated baseline and experiments are missing more recent depth completion baselines (17B6, cTeK); (9) limited evaluation dataset for different sparsity levels and distribution (17B6, cTeK); (10) choice of concatenation vs multiplication (17B6); (11) fairness of comparisons and mismatch in implementations of the methods compared, e.g., Marigold-DC could be evaluated using its original implementation procedure instead of “one out of ten samples.” Likewise, Depth Anything V2 + Least Square originally achieves 4.4 AbsRel on NYUv2, while the reproduced version shows 12.0 AbsRel. (cTeK).

Many of the details asked by the reviewers were lacking in the initial submission, which the authors attributed to the page limit. However, this is indicative of the organization of the content, which was also pointed out by the reviewers. The AC suggests the authors improve the presentation of the manuscript in the next revision.

[A] Wong et al. Unsupervised Depth Completion From Visual Inertial Odometry. RAL 2020.

[B] Tang et al. Bilateral Propagation Network for Depth Completion. CVPR 2024.

**Reviewer Concerns:**

The authors posted a rebuttal to address the following points:

(1) strong concerns about the theoretical section, its validation, and the lack of justification of assumptions made (rCKa, Eiw5): The authors responded to rCKa by "Our decomposition method has demonstrated its effectiveness to a greater extent...natural that you have the concern about the theoretical section. We will include the statistics during training in the latest version of the paper." This does not answer the questions nor address the concerns raised by the reviewers.

(2) unclear and poorly presented content and results in the paper, particularly Figure 1, Tables 3-5 (rCKa, cTeK): The authors mentioned that the lack of description was due to space limit. Nonetheless, the authors should  This point was not addressed.

(3) lack of comparison on the use of learned multiplier and other (naive) outputs, e.g., metric depth (rCKa): The authors did not respond to this point.

(4) the approach is largely systems-based with combination of many existing architectures (rCKa, Eiw5): The authors' response of "Actually our method is not complex. We only replace the DPT head with 3DGS rendering and some necessary modules to regress the gaussian parameters. The interpolation, multiplication operation was routine, and the pipeline is coupled two stage as PromptDA." confirms the reviewers' concern of a largely systems-based approached. This did not address the point raised.

(5) lack of insights and novelty (rCKa, Eiw5): The authors stated difference between their method of other depth completion methods that aligns affine-invariant depth to metric depth, but the AC notes that this is a minor differentiation and does not address the limited scope of novelty pointed out by the reviewers.

(6) possible sensitivity of the interpolation component to noise in sparse points and unrealistic assumptions of sparse point pattern (rCKa, 17B6): The authors state "To against the LiDAR noise, we think it could be better to make use of the rich image prior of pretrained models, e.g. diffusion models with few denoising steps, to maintain consistency of the initial depth map, just as your precious suggestion." This is an unsubstantiated answer and does not address the point raised by the reviewers.

(7) lack of analysis and discussion on computational cost due to the large number components (17B6): The authors noted computational cost in the supplementary materials. This point is addressed.

(8) NLSPN is an outdated baseline and experiments are missing more recent depth completion baselines (17B6, cTeK): The authors did include Marigold-DC in Tables 1 and 2 and will incorporate OMNI-DC and DepthPrompt in later revision.

(9) limited evaluation dataset for different sparsity levels and distributions (17B6, cTeK): The author insists that uniform spatial sampling is sufficient evaluation of generalization of  non-uniform sampling patterns. However, there is no evidence provided. This point was not addressed.

(10) choice of concatenation vs multiplication (17B6): The authors pointed to this experiment in Table 3 and Figure 3. This point is addressed.

(11) fairness of comparisons and mismatch in implementations of the methods compared, e.g., Marigold-DC could be evaluated using its original implementation procedure instead of “one out of ten samples.” Likewise, Depth Anything V2 + Least Square originally achieves 4.4 AbsRel on NYUv2, while the reproduced version shows 12.0 AbsRel. (cTeK): The authors mentioned that they do not have the computational resource to evaluate Marigold-DC based on their proposed protocol and noted that sampling 10, 100, and 1000 examples produced similar results on the Hypersim val set. While the AC sympathizes with the authors, this is not standard evaluation protocol. Separately, the authors pointed the reviewers to a possible misunderstanding of results reported by DepthAnythingV2.

**Reviewer Scores:**

The AC read the reviews and rebuttal. While the authors have responded to some of the concerns of the reviewers, the rebuttal did not fully address the concerns of the reviewers. Many of the concerns were also shared across reviewers. Based on the responses, the AC believes that all reviewers will maintain their negative scores.

---

### Decision · Program_Chairs · 2026-01-26

Reject